# Bioinformatic characterization of ENPEP, the gene encoding a potential cofactor for SARS-CoV-2 infection

Antti Arppo[1], Harlan Barker[1,2,3]*, Seppo Parkkila[1,2]

1 Faculty of Medicine and Health Technology, Tampere University, Tampere, Finland, 2 Department of Clinical Chemistry, Fimlab Laboratories PLC, Tampere University Hospital, Tampere, Finland, 3 Disease Networks Unit, Faculty of Biochemistry and Molecular Medicine, University of Oulu, Oulu, Finland

☯ These authors contributed equally to this work.
* harlan.barker@tuni.fi

**Data Availability Statement:** All relevant data are within the manuscript and its Supporting Information files.

## Abstract

Research on SARS-CoV-2, the viral pathogen that causes COVID-19, has identified angiotensin converting enzyme 2 (ACE2) as the primary viral receptor. Several genes that encode viral cofactors, such as TMPRSS2, NRP1, CTSL, and possibly KIM1, have since been discovered. Glutamyl aminopeptidase (APA), encoded by the gene ENPEP, is another cofactor candidate due to similarities in its biological role and high correlation with ACE2 and other human coronavirus receptors, such as aminopeptidase N (APN) and dipeptidyl peptidase 4 (DPP4). Recent studies have proposed a role for ENPEP as a viral receptor in humans, and ENPEP and ACE2 are both closely involved in the renin-angiotensin-aldosterone system proposed to play an important role in SARS-CoV-2 pathophysiology. We performed bioinformatic analyses using publicly available bulk (>17,000 samples from 49 distinct tissues) and single-cell (>2.5 million cells) RNA-Seq gene expression datasets to evaluate the expression and function of the ENPEP gene. We also investigated age- and sex-related changes in ENPEP expression. Overall, expression of ENPEP was highest in the small intestine enterocyte brush border and the kidney cortex. ENPEP is widely expressed in a subset of vascular smooth muscle cells (likely pericytes) in systemic vasculature, the heart, and the brain. ENPEP is expressed at low levels in the lower respiratory epithelium. In the lung, ENPEP is most highly expressed in para-alveolar fibroblasts. Single-cell data revealed ENPEP expression in a substantial fraction of ependymal cells, a finding not reported before in humans. Age increases ENPEP expression in skeletal muscle and the prostate, while decreasing it in the heart and aorta. Angiogenesis was found to be a central biological function associated with the ENPEP gene. Tissue-specific roles, such as protein digestion and fat metabolism, were also identified in the intestine. In the liver, the gene is linked to the complement system, a connection that has not yet been thoroughly investigated. Expression of ENPEP and ACE2 is strongly correlated in the small intestine and renal cortex. Both overall and in blood vessels, ENPEP and ACE2 have a stronger correlation than many other genes associated with SARS-CoV-2, such as TMPRSS2, CTSL, and NRP1. Possible interaction between glutamyl aminopeptidase and SARS-CoV-2 should be investigated experimentally.

**Funding:** SP received funding from Academy of Finland and Jane and Aatos Erkko Foundation. The funders had no role in study design, data collection and analysis, decision to publish, or preparation of the manuscript.

**Competing interests:** The authors have declared that no competing interests exist.

## Introduction

The coronavirus disease 2019 (COVID-19) pandemic presented the world with a health crisis thus far unprecedented in the 21st century and prompted extensive research into the molecular mechanisms of the underlying pathogen, SARS-CoV-2 [1]. Belonging to the genus Betacoronaviridae, the virus was found to be closely related to the prior SARS-CoV responsible for the SARS epidemic, with evident similarities: both use the same transmembrane enzyme, angiotensin converting enzyme 2 (ACE2), as a viral receptor, a trimeric spike (S) glycoprotein for facilitating viral binding and entry, and the host protease transmembrane serine protease (TMPRSS2) to prime the fusion machinery and achieve cellular entry [2].

However, the resulting diseases are quite different, with COVID-19 characterized by high transmissibility and variable severity, ranging from asymptomatic to acute respiratory distress syndrome (ARDS) and death [3, 4]. Despite the marked genomic and structural overlap between the viruses, differences were found in key viral structures: the SARS-CoV-2 S protein, which is 80% homologous to SARS-CoV, was found to have differences in the receptor binding motif and to possess a novel junctional furin cleavage site between the S1 and S2 subunits [5].

Several studies on the tissue tropism of ACE2, the primary viral receptor, found that it was rather weakly expressed in the lung, contrary to expectations based on the typical manifestation of COVID-19 as a respiratory illness [4], with high expression in organs such as the small intestine, colon, and kidney [6, 7]. In a study by Qi et al., 13 human tissues were analyzed for ACE2 expression, and type II alveolar pneumocytes were found to exhibit 4.7-fold lower ACE2 expression than the average of all ACE2-expressing cells [8]. These findings are further supported by our prior study characterizing ACE2 expression based on bulk and single-cell RNA-Seq datasets, which reported low levels of ACE2 expression in the lung [9].

These discoveries challenged preconceptions on SARS-CoV-2 tissue tropism, with the changes in the S protein further hinting that factors unique to SARS-CoV-2 were at play. This was indeed shown to be the case when Cantuti-Castelvetri et al. demonstrated that neuropilin 1 (NRP1), a transmembrane protein involved in angiogenesis, augments SARS-CoV-2 priming and entry by acting on the S1/S2 furin cleavage site unique to the virus [10]. Currently, many molecules, such as cathepsin L (CTSL), a lysosomal protease that enables host cell invasion after endocytosis [11, 12]; CD147, an extracellular matrix metalloprotease that functions as an independent receptor [13]; and kidney injury molecule 1 (KIM1), a possible secondary viral receptor, have been shown to be utilized by SARS-CoV-2 [14].

Previous studies, such as that of Qi et al., have identified additional potential coreceptors and auxiliary proteins for SARS-CoV-2. Using expression data of known viral receptors and other membrane proteins, ANPEP, ENPEP and DPP4 were identified as the top candidates based on strong coexpression (>0.8) with ACE2 in several tissues [8]—the former two encoding zinc metalloenzymes akin to ACE2 [15, 16]. Notably, the gene products of both ANPEP (aminopeptidase A) and DPP4 (dipeptidyl peptidase 4) are proteases known to act as human coronavirus receptors, interacting with HCoV-229E and MERS-CoV, respectively [17, 18]. Further substantiating these findings, our previous study confirmed high coexpression of ACE2 mRNA with both ENPEP and ANPEP transcripts [9].

The prediction of host protein binding proteins has likewise been a new avenue for preparing for and mitigating the effects of human viral pathogens. Several studies utilizing novel computational tools have predicted the likelihood of human proteins acting as viral receptors. One of these studies utilized a Random Forest (RF) machine learning model to score all cell membrane proteins, identifying 1,424 proteins as comprising the receptorome of the human-infecting virome [19]. Within this set, ENPEP ranks 451 (RF score of 0.6627; scale 0–1), ACE2 ranks 233 (0.7212), ANPEP ranks 39 (0.8616), and DPP4 ranks 28 (0.8758). Similarly, in

another study implementing a generalized boosted machine learning model, the scores of ENPEP (0.356; scale 0–1) and ACE2 (0.369) were similar to those of probable mammalian viral receptors [20].

ENPEP is the gene encoding the protein glutamyl aminopeptidase, also known as aminopeptidase A (APA), human kidney differentiation antigen (gp160) [21] and murine B-lymphocyte differentiation antigen (GP-1/6C3) [22]. Subsequently, we refer to the gene as ENPEP and the protein as APA. A type 2 membrane zinc metalloenzyme, APA is a general-purpose protease that primarily serves to cleave the acidic N-terminal aspartate and glutamyl residues of a variety of peptides, most significantly angiotensin 2 (ANG2, ang1-8), cholecystokinin-8 (CCK8), and neurokinin A [23–26]. The presence of $Ca^{2+}$ augments both the enzymatic efficiency and specificity of APA, increasing its selectivity towards the aforementioned acidic amino acids. Conversely, in the absence of calcium, APA can bind a wider range of N-terminal neutral and basic amino acids, some of which are cleaved, while others bind without proteolysis [24, 27, 28].

APA comprises four major structural features: an N-terminal intracellular tail, a single-pass alpha-helix transmembrane anchor, a stalk segment, and a C-terminal zinc metalloprotease ectodomain. The ectodomain changes conformation, with the binding site alternating between open and closed states to facilitate substrate loading and catalysis, respectively. Like other members of the large zinc metalloprotease family, such as ACE2 and APN, APA typically forms dimeric structures in the cell membrane [27].

APA and ACE2, the primary SARS-CoV-2 receptor, are very closely involved physiologically. Before COVID-19, both proteins were studied mainly for their relationship to the renin-angiotensin-aldosterone system (RAAS), where both cleave angiotensin 2 (ANG2 or ANG1-8), classically regarded as one of the most important pressor metabolites. ACE2 and APA cleave different ends of the peptide: ACE2 cleaves the C-terminal phenylalanine to create (ang1-7), whereas APA cleaves the N-terminal aspartate of ANG2 (ang1-8) to create ANG3 (ang2-8) [23, 29].

Similar to ANG2, ANG3 affects blood pressure by binding to angiotensin type 1 and 2 receptors (AT1 and AT2). This effect appears to be most significant in the central nervous system, where it preferentially binds to AT1 receptors, increasing blood pressure by increasing sympathetic activity, initiating vasopressin release, and modulating the baroreceptor reflex [25]. Conversely, outside the CNS, reduced APA activity has been shown to increase blood pressure in murine models [30]. This is likely due to decreased production of ANG3, which interacts with vasodilatory AT2 receptors in the periphery, and prolonged ANG2 activity, considering that APA has an important role in ANG2 catabolism [31, 32]. Murine studies of the APA inhibitor molecule EC33 further support the role of APA in the CNS regulation of blood pressure, with CNS administration of EC33 reducing blood pressure [33]. Accordingly, APA has significant potential as a drug target, with firibastat—a centrally acting APA inhibitor—currently undergoing phase III clinical trials [34, 35].

The RAAS role of APA is of particular interest, as RAA dysregulation has been proposed to play an important part in SARS-CoV-2 pathophysiology, with impaired ACE2 function being linked to some of the deleterious cardiovascular effects seen in COVID-19 such as increased pulmonary artery pressure and coagulation in swine models [36]. APA might be a significant contributor to these adverse effects, due sharing ANG2 as a substrate with ACE2 [23]. With impaired ACE2 function, ANG2 could instead be preferentially processed by APA.

In a broad range of tissues, including the small intestine, kidney cortex, liver, brain and vasculature, APA plays various general and tissue-specific roles, including activity in angiogenesis, kidney function, the endometrial cycle and implantation [37–40], with corresponding connections to several disease processes, such as a variety of cancers, renal dysfunction,

preeclampsia, and possibly COVID-19 via RAAS dysregulation [39, 41–44]. However, gaps in knowledge remain about the role of APA in many organs, such as the respiratory tract and small intestine, despite the latter being a major locus of ENPEP expression [37]. Interest in the gene is compounded by prior strong correlations between ENPEP and ACE2 [8, 9].

By investigating the cell types expressing the ENPEP gene, its associated biological processes, and its possible correlations with ACE2 and other genes of interest using novel bioinformatic approaches, we hope to elucidate the distribution and function of APA, a possible coreceptor for SARS-CoV-2.

## Methods

### ENPEP mRNA expression

The Genotype-Tissue Expression (GTEx) project (v8) has analyzed gene expression in >17,000 samples of 49 distinct tissues from 838 individuals [45]. These expression data were downloaded (gtexportal.org) as transcripts per million (TPM) values for all genes and aggregated to 31 major tissues/organs of interest. Visualization of ENPEP expression was performed with the Matplotlib [46] and Seaborn [47] Python libraries. For statistical comparison of ENPEP expression, GTEx metadata for donor samples were used to group samples by age (<60 and ≥60) and sex (male and female). Expression of ENPEP in these groups was compared by ANOVA using the SciPy [48] stats module ('f_oneway') for each tissue for which the total number of samples was greater than 20.

### Coexpression and gene ontology enrichment analysis

Coexpression analysis of ENPEP gene expression values was performed against all other annotated genes in the GTEx dataset, within each of 49 tissue groups, by Spearman correlation analysis using the SciPy [48] Python library. Genes satisfying the high correlation threshold cutoff (≥0.70) [49] and a Bonferroni-corrected p-value of 0.05 were used to perform gene ontology enrichment analysis with the g:profiler Python library [50] to identify enriched terms in biological process (BP), molecular function (MF), cellular component (CC), human phenotype (HP), KEGG pathway, and WikiPathways (WP) ontologies. Fisher's exact test enrichment analysis of genes coexpressed with ENPEP (≥0.50; all tissues) in the GTEx dataset was performed using the transcription factor targets (TFT) gene set cataloged in the MSigDB (vers. 2023.2) database [51].

### ENPEP protein expression

The cellular localization of the APA protein was investigated using immunohistochemistry images from the Human Protein Atlas (https://www.proteinatlas.org/) [52]. According to the resource, the samples were stained with a rabbit anti-human polyclonal antibody (Atlas Antibodies Cat#HPA005128, RRID:AB_1844795, Sigma–Aldrich). The images presented in Figs 2 and 3 are from the following donors: small intestine, 41-year-old male; colon, 67-year-old male; kidney, 28-year-old male; liver, 29-year-old female; lung, 49-year-old female; parathyroid, 60-year-old female; adipose, 50-year-old male; and cerebral cortex, 45-year-old male. The Human Protein Atlas included APA immunostaining from two small intestine samples, three colon samples, three liver samples, three lung samples, one parathyroid sample, six adipose samples, and three cerebral cortex samples.

### ENPEP genomic context and regulation

Identification of correlated enhancer RNAs (eRNAs) was performed using custom Python scripts utilizing data from the HeRA database [53]. eRNAs with a strong correlation (≥0.60)

with the ENPEP gene in any tissue were identified, and all other genes strongly correlated with the eRNA in the same tissue were also retrieved. Comparisons between these genes and the genome neighborhood surrounding the ENPEP gene were made and plotted using the ideogram.js library (https://github.com/eweitz/ideogram). Analysis of the ENPEP promoter region was performed using the TFBSfootprinter [54] tool (https://github.com/thirtysix/TFBS_footprinting), which uses transcription-relevant data from several major databases for the prediction of putative transcription factor binding sites (TFBSs). The Ensembl database identifies one protein-coding transcript for the ENPEP gene, which is indicated by the transcript ID ENST00000265162. To identify possible regulators of transcription of this ENPEP transcript and its promoter region, an analysis was performed using the TFBSfootprinter tool, targeting 2,000 base pairs (bp) upstream and 2,000 bp downstream (relative to the TSS) using 575 Jaspar [55] vertebrate nonredundant TF models.

## Single-cell RNA-Seq

To quantify ENPEP expression at the resolution of single cells, existing scRNA-Seq expression count data was obtained from several large analyses hosted on the CellxGene Discover platform [56] (cellxgene.cziscience.com). The scRNA-Seq experiments included data from the Human Brain Cell Atlas v1.0 (888,263 nonneuronal cells of the brain) [57], Heart Cell Atlas V2 (486,134 cells) [58], Gut Cell Atlas (428,469 cells) [59], and the Human Lung Cell Atlas (584,944 cells) [60]. All analyses of the datasets were performed with the SCANPY [61] Python library. The datasets were downloaded as annotated.h5ad files from the CellxGene database, and the raw count data were used in all downstream analyses. The analyses were performed similarly in each instance, with all settings as default except as noted otherwise. Briefly, count data were normalized to 1,000,000 counts per cell ('sc.pp.normalize_total') and then log-normalized ('sc.pp.log1p'); the top 2,000 genes with highly differential expression were identified ('sc.pp.highly_variable_genes': 'flavor = seurat_v3, layer = counts'); principal component analysis was calculated ('sc.pp.pca': 'n_comps = 30, svd_solver = 'arpack''); neighborhood detection was performed ('sc.pp.neighbors': 'n_neighbors = 15, n_pcs = 15'); 2-D graph representation was computed with Uniform Manifold Approximation and Projection (UMAP) ('sc.tl.umap':_'min_dist = 0.001'); and clustering of cells was performed using the Leiden algorithm ('sc.tl.leiden') [62] at various resolutions ranging from 0.1 to 0.5 to identify clusters of biological significance at varying levels of transcriptional coherence. Clusters were subsequently labeled with cell types and tissues based on existing author annotations, scVI [63] automated annotation provided by CellxGene, and manual review of the literature. Expression of ENPEP was subsequently mapped onto UMAP images in each of the datasets.

From the Human Protein Atlas, raw count and normalized expression (nTPM) scRNA-Seq data was downloaded for 689,601 cells across 31 tissues comprising 557 annotated cell type clusters and 85 distinct cell types. Using custom Python scripts, the fraction of cells expressing ENPEP was computed for each of the 557 cell type clusters and combined with nTPM values.

## Results

### ENPEP mRNA expression is highest in the small intestine and kidney

ENPEP mRNA expression in different tissues was investigated using the publicly available GTEx dataset, and immunohistochemical staining images from the Human Protein Atlas were used to determine how APA, the protein encoded by ENPEP, localizes within tissues of interest. ENPEP mRNA is expressed in a variety of tissues, with the small intestine and kidney cortex exhibiting the highest overall levels of expression. Lower levels of expression were found in visceral adipose tissue, coronary arteries, the lung, and the spleen. Among the sex-based tissues

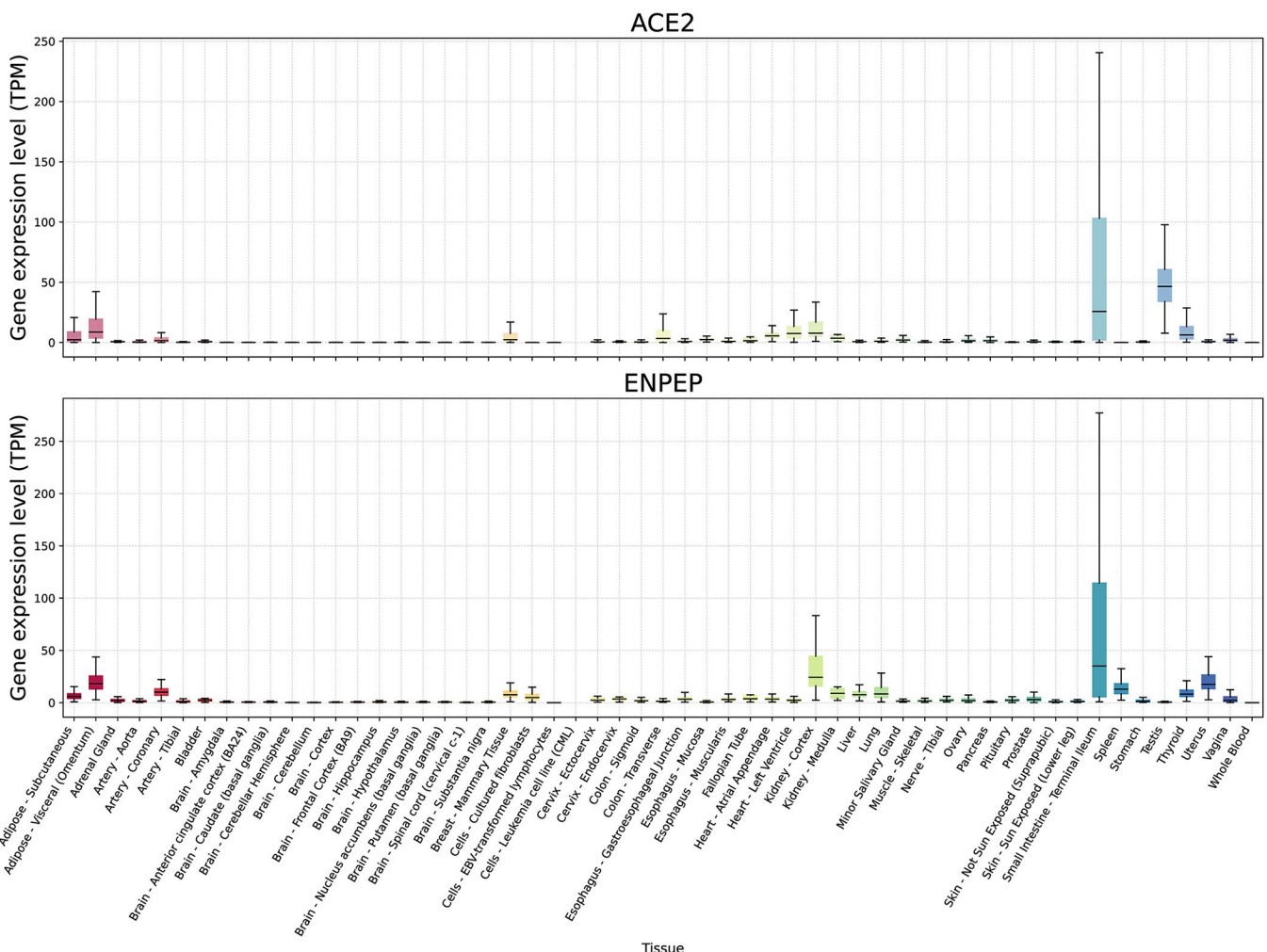

**Fig 1. Tissue specific expression of ACE2 and ENPEP in 55 human tissues.** Bulk RNA-Seq expression data from the GTEx dataset (v8) [45] was retrieved as TPM values and boxplot figures were generated using the Seaborn [47] and Matplotlib [46] Python libraries. Each boxplot displays the distribution of expression values for the gene-tissue intersection: the box represents the interquartile range (IQR), the lower boundary marks the 1st quartile (Q1), the upper boundary marks the 3rd quartile (Q3), and the horizontal line within the box indicates the median. The whiskers extend to the maximum and minimum values within 1.5 times the IQR.

examined, the uterus exhibited the highest levels of ENPEP expression. Overall, ENPEP mRNA expression appears fairly tissue specific, as the majority of its expression is localized in a few select tissues (Fig 1).

APA immunostaining specimens are presented in Figs 2 and 3. Immunostaining in the small intestine and kidney seemed to correlate with the mRNA expression values in the GTEx dataset. Strong signals were found in the apical membrane of enterocytes in the small intestine, whereas staining in enterocytes of the colon was much weaker and intracellular. In the kidney, staining was observed in intraglomerular cells, probably representing podocytes, as well as in Bowman's capsule and proximal tubule epithelium (Fig 2). Prominent immunostaining was also observed in the liver, lung and parathyroid gland (Fig 2). Despite the low overall ENPEP mRNA levels indicated in the GTEx dataset, clear signals were found in the sinusoidal lining cells of the liver, hepatocytes, lung alveoli, and vascular structures (Fig 2). The positive alveolar signal of the lung samples could represent pneumocytes or other para-alveolar cells not

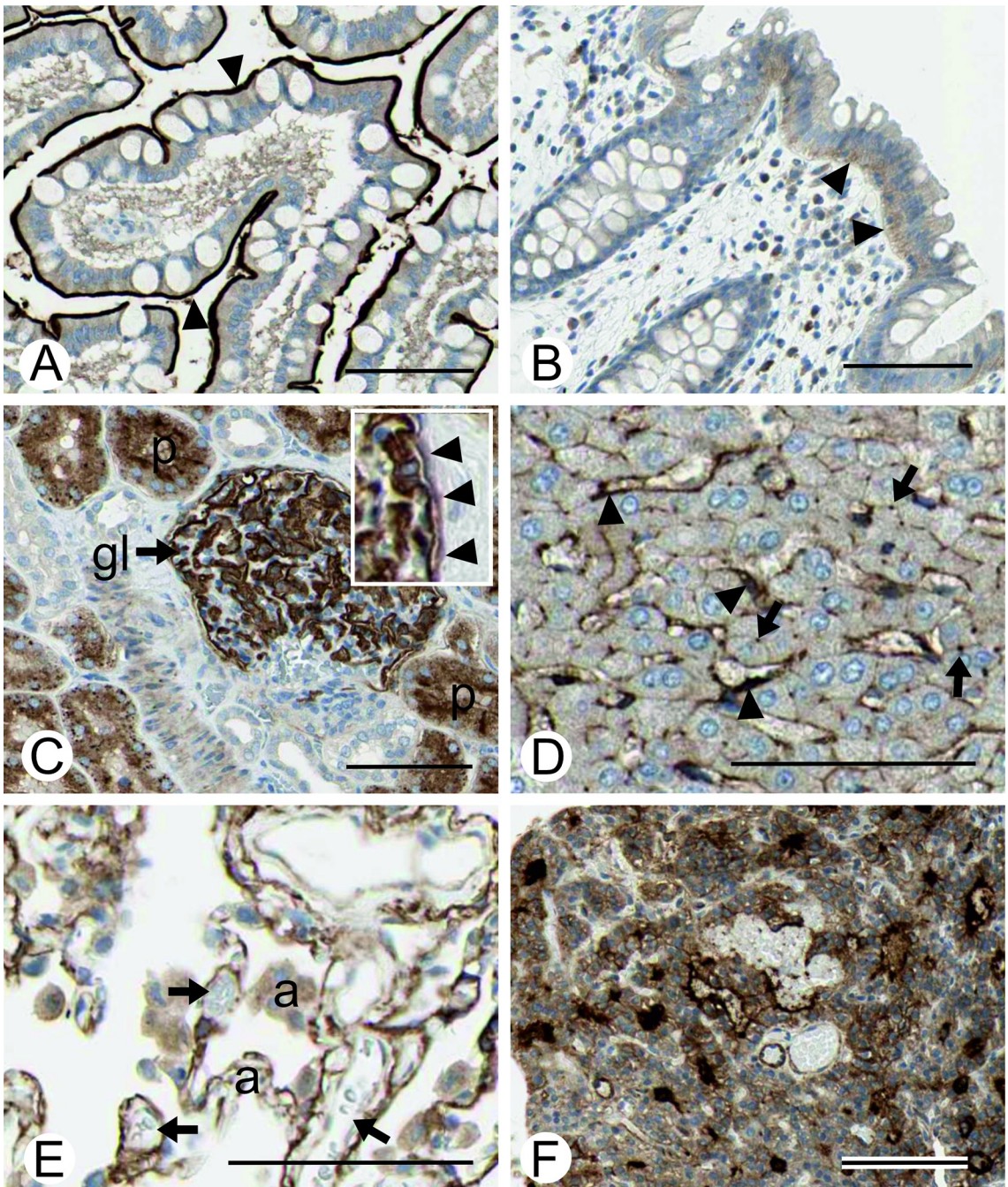

**Fig 2.** Immunohistochemical staining of APA in the human small intestine (A), colon (B), kidney (C), liver (D), lung (E), and parathyroid gland (F). In the small intestine, the strong positive reaction is confined to the apical brush border of enterocytes (arrowheads). The colon shows only a very faint intracellular signal, which is visible in the basal part of the enterocytes (arrowheads). In the kidney, strong positive immunoreactivity is present in both the glomerulus (gl) and proximal tubules (p). The positive signal in the glomerulus shows a typical podocyte-type staining pattern, and the signal is also present in Bowman's capsule epithelium (arrowheads in insert). In the liver, immunoreactions are prominent in sinusoidal lining cells (arrowheads). A positive signal is also present at the plasma membranes of hepatocytes. Arrows indicate the location of the canalicular (hepatocyte apical) membrane showing an intense punctate signal. In the lung, positive staining is localized in alveolar cells (a). The cells surrounding small blood vessels (arrows) also showed immunoreactivity. The parathyroid gland shows a strong signal in the glandular cells. Immunostaining images were produced from scanned tissue specimens of the Human Protein Atlas (https://www.proteinatlas.org/) [52]. Bars 100 μm.

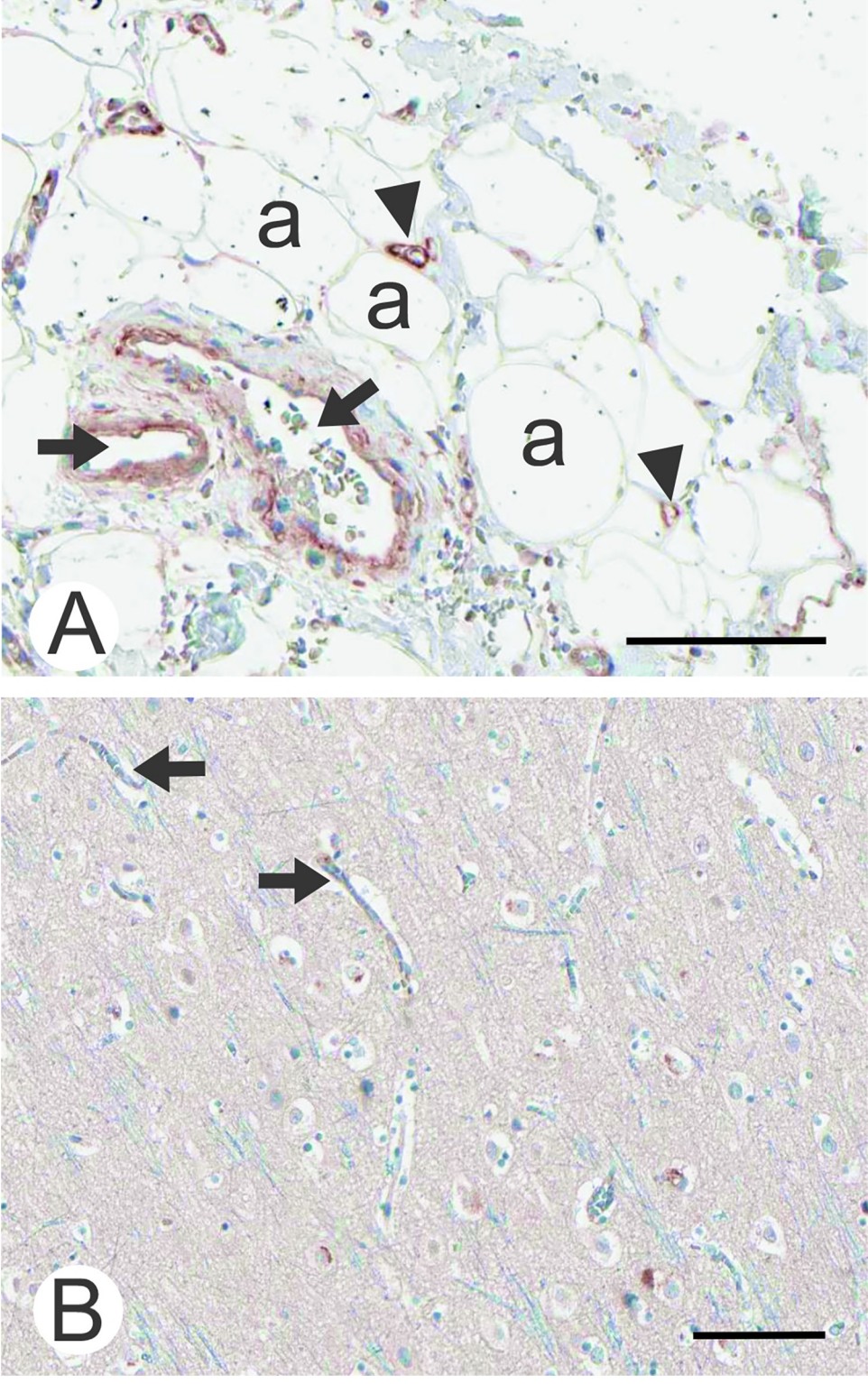

**Fig 3.** Immunohistochemical staining of APA in human adipose tissue (A) and the cerebral cortex (B) focusing on microvasculature. The positive signal of adipose tissue is mainly located in blood vessel walls, including capillaries. Arterioles (arrows) and capillaries (arrowheads) are indicated in the image. Adipocytes (a) are negative for immunostaining. In the brain, capillary beds (arrows) are negative or only faintly positive. Immunostaining images were produced from scanned tissue specimens of the Human Protein Atlas (https://www.proteinatlas.org/) [52]. Bars 100 μm.

differentiable with light microscopy, while the vascular signals could represent expression in endothelial cells, pericytes and/or fibroblasts. In addition to the lung, vascular staining was also clearly visible in adipose tissue, with staining in both capillaries and arterioles. However, vascular staining in the cerebral cortex was very faint or negative (Fig 3).

## ENPEP expression in single-cell data

As immunohistochemistry cannot be used to reliably quantify gene expression in distinct cell populations, single-cell RNA-Seq expression data from the Human Protein Atlas, comprising 85 cell types across 31 was used to further investigate ENPEP expression within tissues. The Human Protein Atlas data was complemented with large tissue-specific datasets from the lung, brain, heart, and small intestine.

Table 1 presents all cell populations in the Human Protein Atlas dataset that had an ENPEP prevalence greater than 30% and the top 10 most upregulated differentially expressed genes (DEGs) for each cell cluster. Among all samples, proximal enterocytes of the small intestine exhibited the highest prevalence of ENPEP expression, with different enterocyte populations comprising the top three cell populations by expression level (>575 nTPM) and prevalence of expression (>79% of cells). Other tissue-specific cell types exhibiting prevalent ENPEP expression included kidney proximal tubular cells (47%), liver hepatocytes (41%), and brain excitatory neurons (36%). ENPEP expression was prevalent in the smooth muscle cell populations of many tissues, such as vascular tissue, the placenta, the thymus, adipose tissue, and the ovary (Table 1). This difference appeared to be pronounced in vascular tissue, which revealed a >50% prevalence of ENPEP expression in two separate smooth muscle cell populations (Table 1).

Notably, only certain clusters of these smooth muscle cells expressed ENPEP rather than smooth muscle cells in general (S1 Table). The individual smooth muscle cell clusters with the highest ENPEP expression disproportionately expressed pericyte marker genes defined in the CellxGene database [56] (e.g., NOTCH3, PDGFRB, MCAM, IGFBP7, STEAP4, and TAGLN) compared to other cell clusters, likely indicating that most of these smooth muscle cell populations are in fact pericytes rather than vascular smooth muscle cells. Because pericytes are not separately annotated in the Human Protein Atlas data, they are likely conflated into the smooth muscle cell category. NOTCH3 in particular was common among the marker genes, as it was among the top 15 upregulated DEGs of every smooth muscle cell population with more than 10% ENPEP prevalence, except in placenta samples (S1 Table). Despite prevalent ENPEP expression in several vascular cell clusters, ENPEP expression was almost absent in most endothelial cells, with the exception of a single, small endothelial cell cluster in skin tissue with 15.5% prevalence and 25.3 nTPM.

In addition to smooth muscle and enterocytes, a variety of organs, such as kidney proximal tubular cells (47%), hepatocytes (41%), and brain excitatory neurons (36%), had cell populations with an ENPEP prevalence >30%, with Table 1 showing all 24 of these cell clusters. However, the hepatocyte population with the highest ENPEP expression was a small cluster of only 30 cells, with a prevalence of approximately 8% in some of the other larger hepatocyte populations (S2 Table). Several placental cytotrophoblast populations in the dataset had a marked prevalence of expression (>30%), with the highest at 59% prevalence and 89 nTPM (Table 1).

The main immune cell populations labeled in the Human Protein Atlas data—B cells, T cells, NK cells, plasma cells, macrophages, dendritic cells, neutrophils, monocytes, and mixed immune cells—had only minimal ENPEP expression, with the highest prevalence in placental mixed immune cells at only 2.1% (S3 Table). No significant expression was found in the immune-related tissues of bone marrow, lymph nodes, or spleen. An exception to this is the

**Table 1. Normalized ENPEP expression (nTPM) in single-cell mRNA data from the Human Protein Atlas (https://www.proteinatlas.org/) [64].**

| Tissue | Cell type | ClusterID | ENPEP expressing cells in cluster | Total cells in cluster | Prevalence of expression, % | Expression level, nTPM | Top ten upregulated differentially expressed genes |
|---|---|---|---|---|---|---|---|
| Small intestine | Proximal enterocytes | 0 | 816 | 852 | 95.77 | 730.8 | PHGR1, ADIRF, ACE, RTN4, C3orf85, ANPEP, ATP1A1, GUCA2A, TDP2, FTH1 |
| Small intestine | Proximal enterocytes | 1 | 665 | 820 | 81.1 | 698.3 | CD74, HLA-B, PRAP1, FABP6, HLA-DRA, DPEP1, HLA-DRB5, CDHR5, IL32 |
| Small intestine | Proximal enterocytes | 5 | 469 | 592 | 79.22 | 575.6 | GUCA2A, SELENOP, ADIRF, GPX4, PRAP1, SERPINA1, DPEP1, CDHR5, PHGR1, CYP3A4 |
| Small intestine | Undifferentiated cells | 3 | 499 | 711 | 70.18 | 137.7 | SPINK1, UQCRH, REG1A, EEF1A1, CCL25, NACA, SNRPD2, RACK1, TMSB10, ATP5MC2 |
| Small intestine | Proximal enterocytes | 4 | 398 | 626 | 63.58 | 388 | FABP6, FABP1, RBP2, FAM151A, KHK, CRIP1, BLVRB, FUOM, REEP6, NFKBIA |
| Vascular | Smooth muscle cells | 17 | 115 | 192 | 59.9 | 107.9 | NOTCH3, TBX2, BCAM, HES4, TINAGL1, GUCY1A1, LBH, MCAM, KCNAB1, CRIP1 |
| Placenta | cytotrophoblasts | 12 | 430 | 731 | 58.82 | 89 | SIGLEC6, DUSP9, SLC43A2, SERINC2, EFEMP1, PAGE4, MEST, FAM118A, IFI6, SLC22A11 |
| small intestine | Undifferentiated cells | 6 | 191 | 346 | 55.2 | 72.6 | EIF3E, NPM1, HNRNPA1, EEF1A1, NACA, RACK1, PPP1R1B, PABPC1, ACTG1, SLC25A6 |
| Vascular | Smooth muscle cells | 6 | 298 | 556 | 53.6 | 94.7 | NOTCH3, STEAP4, TINAGL1, GGT5, PARM1, ADGRF5, INPP4B, FHL5, ISYNA1, NR2F2 |
| Kidney | Proximal tubular cells | 9 | 242 | 510 | 47.45 | 157 | POLR2J3, N4BP2L2, ACSM2A, ACSM2B, VMP1, DDX5, DDX17, RNF213, PNISR, SYNE2 |
| Liver | Hepatocytes | 18 | 26 | 64 | 40.62 | 47.3 | SERPING1, CYP2C8, C4BPA, C9, C1S, TPT1, FGL1, CYP2C9, SELENOP, CFH |
| Placenta | Cytotrophoblasts | 3 | 672 | 1701 | 39.51 | 38.7 | PAGE4, STMN1, HMGB1, PEG10, HMGN2, GGCT, H2AZ1, SPINT2, MPC2, TSPAN13 |
| Placenta | Smooth muscle cells | 18 | 106 | 275 | 38.55 | 61.5 | C1S, SERPING1, C1R, CFD, PCOLCE, SERPINF1, IFITM3, JUN, SELENOM, IFITM2 |
| Kidney | Proximal tubular cells | 4 | 793 | 2127 | 37.28 | 305.4 | ITM2B, CD63, NAT8, PCSK1N, LAPTM4A, GSTP1, TNFSF10, CLTRN, TSPAN1, APP |
| Thymus | Smooth muscle cells | 12 | 205 | 566 | 36.22 | 89.5 | STEAP4, NOTCH3, GGT5, TIMP3, TAGLN, TINAGL1, TPM1, ADGRF5, EPS8, PDGFRB |
| Brain | Excitatory neurons | 32 | 297 | 826 | 35.96 | 22 | POU6F2, CDH12, LAMA2, PTPRD, RIT2, SNTG1, NRG1, LDB2, FOXP1, KHDRBS2 |
| Adipose tissue | Smooth muscle cells | 15 | 415 | 1197 | 34.67 | 153.4 | IGFBP7, TINAGL1, SPARCL1, ADIRF, NOTCH3, CALD1, MYL9, TPM1, TAGLN, EPS8 |
| Placenta | cytotrophoblasts | 13 | 239 | 725 | 32.97 | 55.6 | PAGE4, IFI6, FAM3B, XAGE3, FXYD3, ISYNA1, CYSTM1, VGLL1, SMAGP, EFEMP1 |
| Thymus | Smooth muscle cells | 10 | 197 | 614 | 32.08 | 67.2 | IGFBP7, NOTCH3, BCAM, C11orf96, TINAGL1, MFGE8, MGST3, SOD3, C1QTNF1, ADIRF |

*(Continued)*

**Table 1.** (Continued)

| Tissue | Cell type | ClusterID | ENPEP expressing cells in cluster | Total cells in cluster | Prevalence of expression, % | Expression level, nTPM | Top ten upregulated differentially expressed genes |
|---|---|---|---|---|---|---|---|
| Endometrium | Endometrial stromal cells | 14 | 443 | 1401 | 31.62 | 73.4 | COL6A2, APOE, MMP2, PLAGL1, TIMP2, AEBP1, COL1A1, CPXM1, CRISPLD2, FBLN2 |
| Placenta | cytotrophoblasts | 0 | 1340 | 4365 | 30.7 | 38.5 | PAGE4, DUSP9, BCAM, SPINT1, ACSS1, PEG10, SMAGP, EFEMP1, MORC4, SLC22A11 |
| Brain | Excitatory neurons | 16 | 588 | 1927 | 30.51 | 17.3 | CTNNA2, RXFP1, RALYL, NAV3, PDE4D, CADPS2, HTR1F, SYT1, NELL2, CPNE4 |
| Testis | Sertoli cells | 19 | 13 | 43 | 30.23 | 62.6 | LGALS1, SPARC, CPE, MAP1B, PPP1R14A, MGST3, SH3BGRL, MT2A, ITM2C, CALD1 |
| Ovary | Smooth muscle cells | 11 | 541 | 1792 | 30.19 | 108.6 | RGS5, HIGD1B, PHLDA1, IGFBP7, CPE, TINAGL1, PDGFRB, COX4I2, NOTCH3, SEPTIN7 |

thymus, where two smooth muscle cell populations had a prevalence of expression >30%, along with some fibroblast expression of ENPEP (S3 Table).

The Human Protein Atlas included scRNA-Seq data from the lung and bronchus (S4 Table). In the lung, 6.5% of fibroblasts (25.7 nTPM) and 6.5% (18.8 nTPM) of smooth muscle cells express ENPEP. These smooth muscle cells are likely pericytes based on some of the top upregulated marker genes (PDGFRB, CALD1, NR2F2, MCAM, TINAGLI1, and NOTCH3). Other cell types, such as ciliated cells, alveolar cells, and macrophages, had significantly lower prevalence (<0.43%). Every surface cell type of human bronchial tissue appeared to have low ENPEP expression (S4 Table).

Human lower respiratory tract expression was also investigated using the Human Lung Atlas dataset (Fig 4). Similar to the Human Protein Atlas data, epithelial expression of ENPEP was low in all parts of the respiratory tract, with no marked expression in the trachea or bronchi and expression in only individual cells of the alveolar epithelium (type 1 and type 2

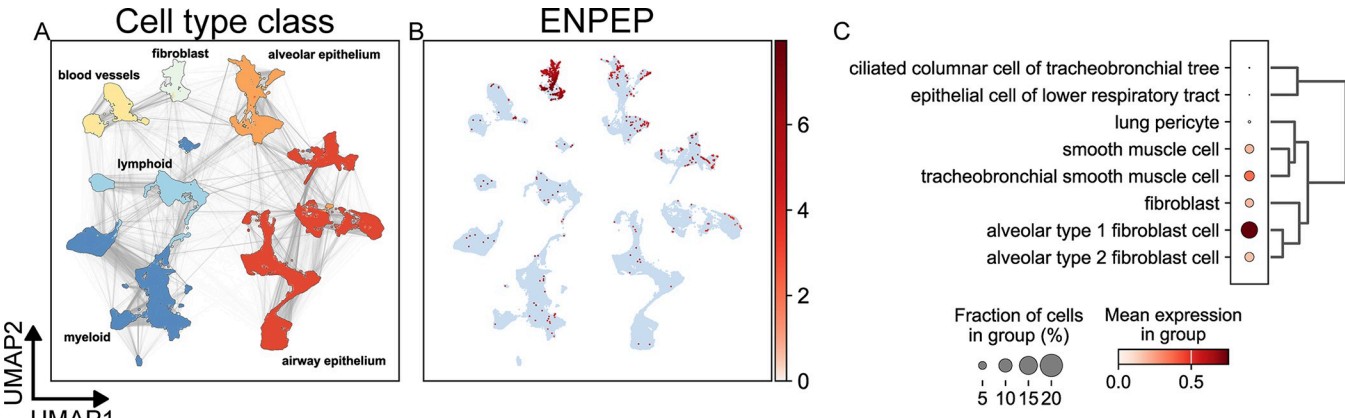

**Fig 4. scRNA-Seq analysis of human lung tissue.** Data from the Human Lung Cell Atlas (584,944 cells) [60] was retrieved from the CellxGene [56] data platform. UMAP plots depicting cell types (A) and expression of the ENPEP gene (B) were calculated and generated using the SCANPY Python library (vers. 1.9.3) [61]. (C) A dot plot depicting the expression of ENPEP by cell type cluster shows the fraction of cells in the cell type expressing ENPEP and the mean of normalized expression of all cells in that cell type cluster. Cell type annotations are a combination of author cell type annotations, scVI [63] automatic annotation performed by the CellxGene platform, and manual review of the literature.

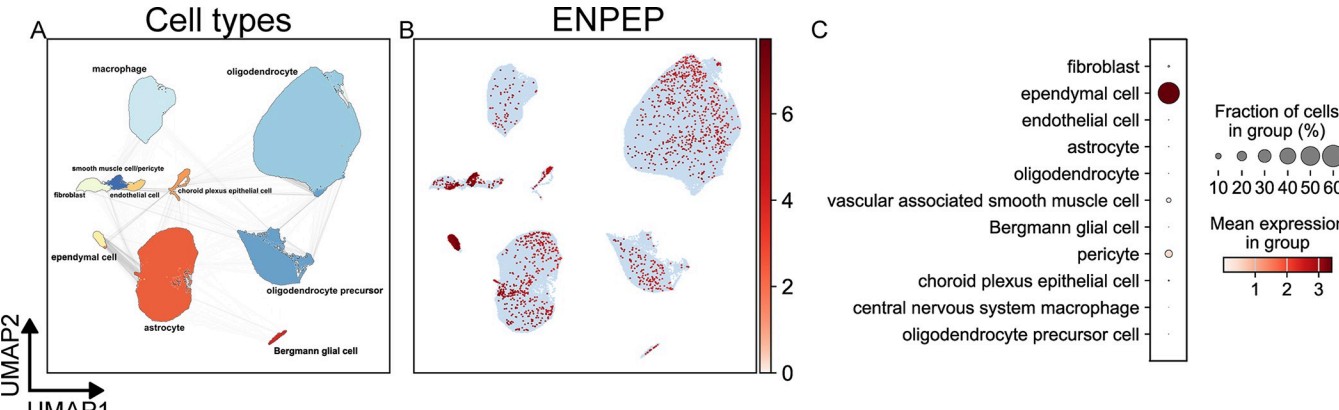

**Fig 5. scRNA-Seq analysis of human brain tissue.** Data from the Human Brain Cell Atlas (v1.0; 888,263 nonneuronal cells of the brain) [57] was retrieved from the CellxGene [56] data platform. UMAP plots depicting cell types (A) and expression of the ENPEP gene (B) were calculated and generated using the SCANPY Python library (vers. 1.9.3) [61]. (C) A dot plot depicting the expression of ENPEP by cell type cluster shows the fraction of cells in the cell type expressing ENPEP and the mean of normalized expression of all cells in that cell type cluster. Cell type annotations are a combination of author cell type annotations, scVI [63] automatic annotation performed by the CellxGene platform, and manual review of the literature.

pneumocytes). Lung expression of ENPEP mainly localized to several types of fibroblast and smooth muscle cell populations, most prevalently alveolar fibroblasts, which are cells that inhabit the space between pneumocytes and the vascular endothelium. Tracheobronchial smooth muscle cells and unspecified smooth muscle cells also exhibited a prevalence of expression between 5–10%. No cell type exhibited a prevalence of expression greater than 20% in the lower respiratory tract (Fig 4).

In the Human Protein Atlas data of the brain, ENPEP was expressed in select neuron populations, both excitatory and inhibitory (S5 Table). A total of 10 neuron populations were found to have a prevalence of expression >10%, with the top five ranging from 19–36% in prevalence (17.9–22 nTPM). Most neuron populations had no significant ENPEP expression. Aside from neurons, the other brain parenchymal cell types, such as astrocytes, microglia and oligodendrocytes, exhibited almost no ENPEP expression (S5 Table).

The Human Brain Cell Atlas dataset was used to evaluate ENPEP expression in nonneuronal cells of the brain in more detail, as shown in Fig 5. Surprisingly, the majority of ependymal cells exhibited ENPEP expression—a finding not observed in the Human Protein Atlas data. However, ependymal cells were not separately annotated in the Human Protein Atlas dataset and are not present throughout the brain, likely explaining the discrepancy. Aside from ependymal cells, other glial cell populations did not exhibit meaningful ENPEP expression. The other groups with notable ENPEP expression were vascular cell types, with a prevalence of expression between 10 and 20% in both pericytes and smooth muscle cells (Fig 5).

Analysis of data from the Human Protein Atlas of the tongue, esophagus, stomach, small intestine, colon, and rectum revealed that gastrointestinal expression of ENPEP occurs mainly in the enterocytes of the small intestine, with low epithelial expression in all other parts of the GI tract (S6 Table). Similarly, small intestine data from the Gut Cell Atlas indicated that enterocytes were the standout cell type of ENPEP expression: most enterocyte populations had an ENPEP prevalence greater than 50% (Fig 6). Some expression was also observed in Paneth cells, other epithelial cells, and pericytes (Fig 6).

The Human Protein Atlas contained only 9 cell clusters from the heart, where ENPEP was expressed primarily in smooth muscle cells (9.9% prevalence, 67.6 nTPM) (S7 Table). Due to the prevalent vascular expression of ENPEP in data from the Human Protein Atlas (Table 1),

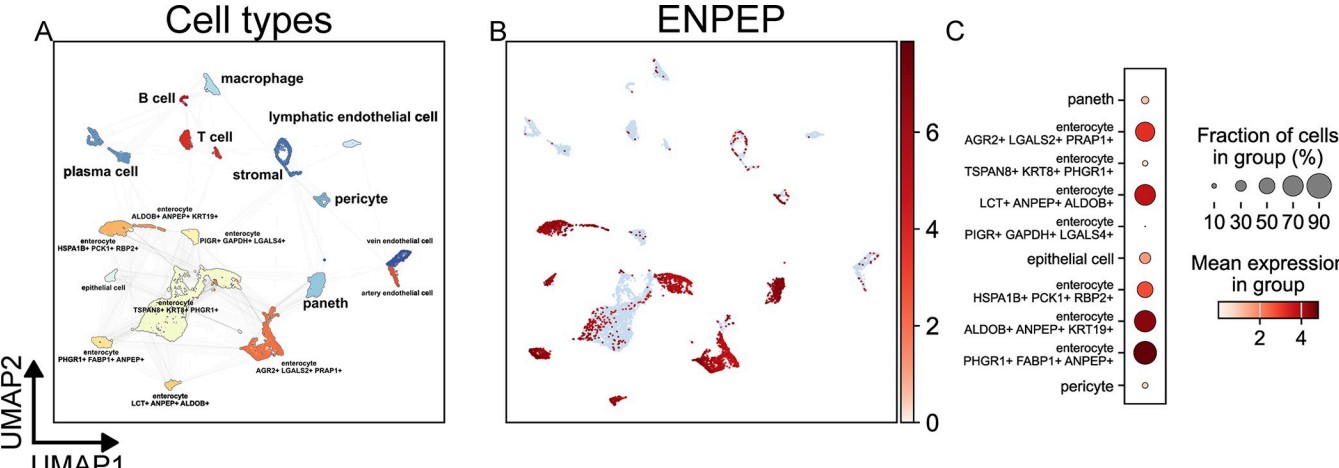

**Fig 6. scRNA-Seq analysis of the small intestine.** Data from the Gut Cell Atlas (428,469 cells) [59] was retrieved from the CellxGene [56] data platform. The data was filtered to cells of healthy adult small intestine, resulting in a final cell count of 23,890. UMAP plots depicting cell types (A) and expression of the ENPEP gene (B) were calculated and generated using the SCANPY Python library (vers. 1.9.3) [61]. (C) A dot plot depicting the expression of ENPEP by cell type cluster shows the fraction of cells in the cell type expressing ENPEP and the mean of normalized expression of all cells in that cell type cluster. Cell type annotations are a combination of author cell type annotations, scVI [63] automatic annotation performed by the CellxGene platform, and manual review of the literature.

we investigated ENPEP expression in the heart with data from the Heart Cell Atlas (Fig 7). Single-cell analysis of the human heart revealed three cell populations with a high prevalence and relatively high mean expression of ENPEP: pericytes, smooth muscle cells, and a small cardiomyocyte population separate from the larger atrial and ventricular cardiomyocyte populations. In contrast to those in the lung, fibroblasts in the heart did not exhibit notable ENPEP expression. Pericytes and smooth muscle cells of the heart were found to express ENPEP, whereas vascular endothelial cells were mostly absent of ENPEP expression—findings similar to brain single-cell data (Fig 7).

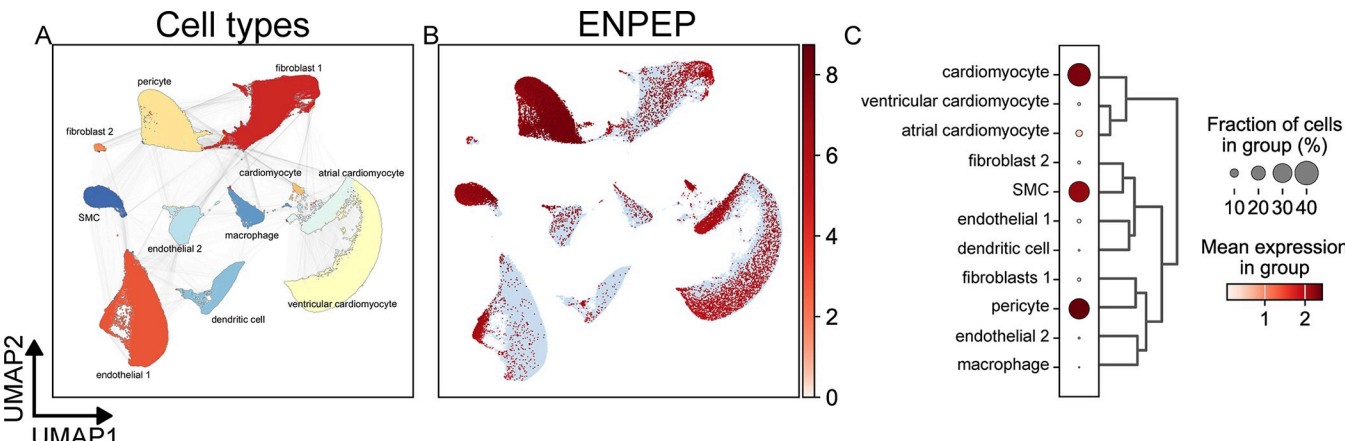

**Fig 7. scRNA-Seq analysis of ENPEP expression in the human heart.** Data from the Heart Cell Atlas (V2; 486,134 cells) [58] was retrieved from the CellxGene [56] data platform. UMAP plots depicting cell types (A) and expression of the ENPEP gene (B) were calculated and generated using the SCANPY Python library (vers. 1.9.3) [61]. (C) A dot plot depicting the expression of ENPEP by cell type cluster shows the fraction of cells in the cell type expressing ENPEP and the mean of normalized expression of all cells in that cell type cluster. Cell type annotations are a combination of author cell type annotations, scVI [63] automatic annotation performed by the CellxGene platform, and manual review of the literature.

## ENPEP mRNA expression does not vary with age in the lung, intestine or kidney

ENPEP mRNA expression in various tissues was further investigated in relation to age, an important prognostic factor for COVID-19 (Fig 8). When comparing ENPEP expression between 25 and 75 years of age and using age brackets of 10 years, statistically significant differences in ENPEP expression between age groups were not found in the lung, intestine, or kidney. A significant difference was found in 12 tissues in the GTEx dataset, including skeletal muscle (p = 3.11 x $10^{-11}$), prostate (p = 4.84 x $10^{-4}$), liver (p = 9.19 x $10^{-4}$), cerebellar hemisphere (p = 0.009199), esophageal mucosa (p = 1.04 x $10^{-2}$), substantia nigra (p = 1.04 x $10^{-2}$), vagina (p = 2.23 x $10^{-2}$), tibial nerve (p = 2.29 x $10^{-2}$), gastroesophageal junction (p = 2.92 x $10^{-2}$), aorta (p = 3.57 x $10^{-2}$), left ventricle of the heart (p = 3.90 x $10^{-2}$), and mammary tissue (p = 4.28 x $10^{-2}$). However, ENPEP mRNA levels did not uniformly increase or decrease with age in these tissues. ENPEP expression increased with age in skeletal muscle, the prostate, and the kidney cortex but decreased in the aorta, left ventricle, and tibial nerve. On the other hand, the three tissues that displayed the highest ENPEP expression—the terminal ileum, kidney cortex, and visceral adipose tissue from the omentum—did not exhibit statistically significant differences in ENPEP mRNA expression between age groups (Fig 8).

The results from a binary comparison of ENPEP expression by age—derived from dividing the samples into two groups, those obtained from subjects aged <60 and those obtained from subjects aged ≥60—did not differ significantly from the ANOVAs performed with narrower age brackets, with all statistically significant findings represented in Table 2. Between the two age groups, significant differences were observed in skeletal muscle (p = 8.77 x $10^{-12}$), prostate (p = 1.57 x $10^{-6}$), liver (p = 1.34 x $10^{-3}$), cerebellar hemispheres (p = 1.73 x $10^{-3}$), esophageal mucosa (p = 8.55 x $10^{-2}$), transverse colon (p = 9.07 x $10^{-3}$), mammary tissue (p = 1.04 x $10^{-2}$), cultured fibroblasts (p = 1.14 x $10^{-3}$), the muscular layer of the esophagus (p = 1.86 x $10^{-2}$), the substantia nigra (p = 2.02 x $10^{-2}$), and the vagina (3.15 x $10^{-2}$) (Table 2).

## ENPEP mRNA expression levels overlap significantly between sexes

A total of 11 tissues (not sex-related) displayed a significant difference in ENPEP expression between the sexes: frontal cortex, pituitary, adrenal gland, visceral adipose (omentum) (20.9 ±9.8 TPM in males, 18.5±9.3 TPM in females, p = 0.007), skeletal muscle (1.8±1.2 TPM in males, 18.5±9.3 in females, p = 0.017), nucleus accumbens, amygdala, sun-exposed skin (lower leg), putamen, tibial artery (1.4±1.3 TPM in males, 1.2±1.2 TPM in females, p = 0.038), and whole blood (0.021±0.027 TPM in males, 0.029±0.078 TPM in females, p = 0.043) (Fig 9). Even in these tissues, ENPEP expression overlapped significantly between sexes, and expression levels were not systematically higher in males or females (Fig 9).

## ENPEP expression correlates with the expression of genes related to angiogenesis, including NRP1

Gene correlation analysis was used to identify related genes from the GTEx dataset bulk RNA-seq data in all samples from all tissues, and in selected tissues. When taking into account all samples, 20 genes had a strong correlation (≥0.70) (S8 Table), the top ten of which are presented in Table 3. The most common associated features were related to angiogenesis, endothelial permeability, and cell migration. Notably, NRP1, encoding a known SARS-CoV-2 cofactor, had the strongest overall correlation with ENPEP (0.80), followed by ADGRL4 (adhesion G protein-coupled receptor L4), TFPI (tissue factor pathway inhibitor), and VEGFC (vascular endothelial growth factor C) (Table 3). Both ADGRL4 and VEGFC are associated with

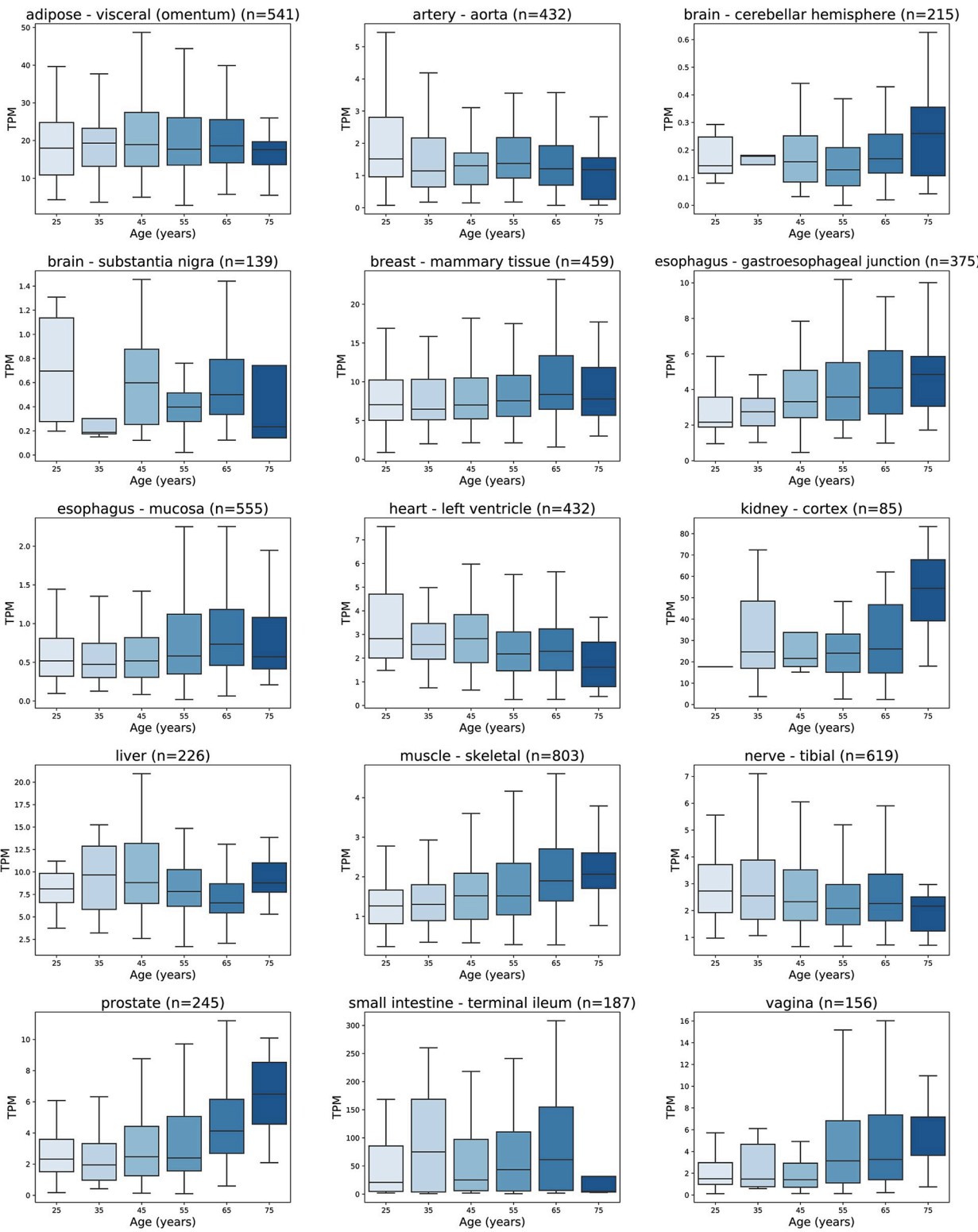

**Fig 8. ENPEP mRNA expression in TPM in the tissues with a statistical significance between age groups (ANOVA; p-value ≤0.05), and the three tissues with the highest ENPEP mRNA expression (terminal ileum, kidney cortex, and visceral adipose tissue) were also significantly different.** Each boxplot displays the distribution of expression values for the gene-tissue intersection: the box represents the interquartile range (IQR), the lower boundary marks the 1st quartile (Q1), the upper boundary marks the 3rd quartile (Q3), and the horizontal line within the box indicates the median. The whiskers extend to the maximum and minimum values within 1.5 times the IQR. Data derived from the GTEx dataset (v8) (https://www.gtexportal.org/home/) [45].

**Table 2. Variation in ENPEP expression by tissue and age group.**

| Tissue | f-value | p-value | TPM in subjects aged <60 | TPM in subjects aged ≥60 |
|---|---|---|---|---|
| Muscle—skeletal | 48.0 | 8.8E-12 | 1.68±1.03 (n = 511) | 2.28±1.44 (n = 292) |
| Prostate | 24.2 | 1.6E-06 | 3.19±2.61 (n = 165) | 5.17±3.52 (n = 80) |
| Liver | 10.6 | 1.3E-03 | 9.50±5.38 (n = 142) | 7.43±2.88 (n = 84) |
| Brain—cerebellar hemisphere | 10.1 | 1.7E-03 | 0.16±0.12 (n = 96) | 0.24±0.22 (n = 119) |
| Esophagus—mucosa | 7.0 | 8.6E-03 | 0.76±0.69 (n = 389) | 0.94±0.75 (n = 166) |
| Colon—transverse | 6.9 | 0.009 | 1.74±3.65 (n = 301) | 2.79±3.23 (n = 105) |
| Breast—mammary tissue | 6.6 | 0.010 | 8.82±5.63 (n = 304) | 10.28±6.01 (n = 155) |
| Cells—cultured fibroblasts | 6.4 | 0.011 | 5.89±4.86 (n = 342) | 7.18±6.17 (n = 162) |
| Esophagus—muscularis | 5.6 | 0.019 | 3.62±2.46 (n = 372) | 4.18±2.28 (n = 143) |
| Brain—substantia nigra | 5.5 | 0.020 | 0.50±0.35 (n = 58) | 0.72±0.64 (n = 81) |
| Vagina | 4.7 | 0.031 | 3.50±3.98 (n = 114) | 5.10±4.28 (n = 42) |

endothelial function and proliferation, and TFPI is a central component of the coagulation cascade, reflecting the role of ENPEP in the vascular system.

The results differed when comparing correlations within separate tissues. Table 3 presents the genes most strongly correlated with ENPEP in selected tissues. Strong gene correlations with coefficients ≥0.70 were present in many tissues of interest, such as the small intestine, colon, kidney, and liver. In the small intestine, 28 genes had a near-perfect correlation with

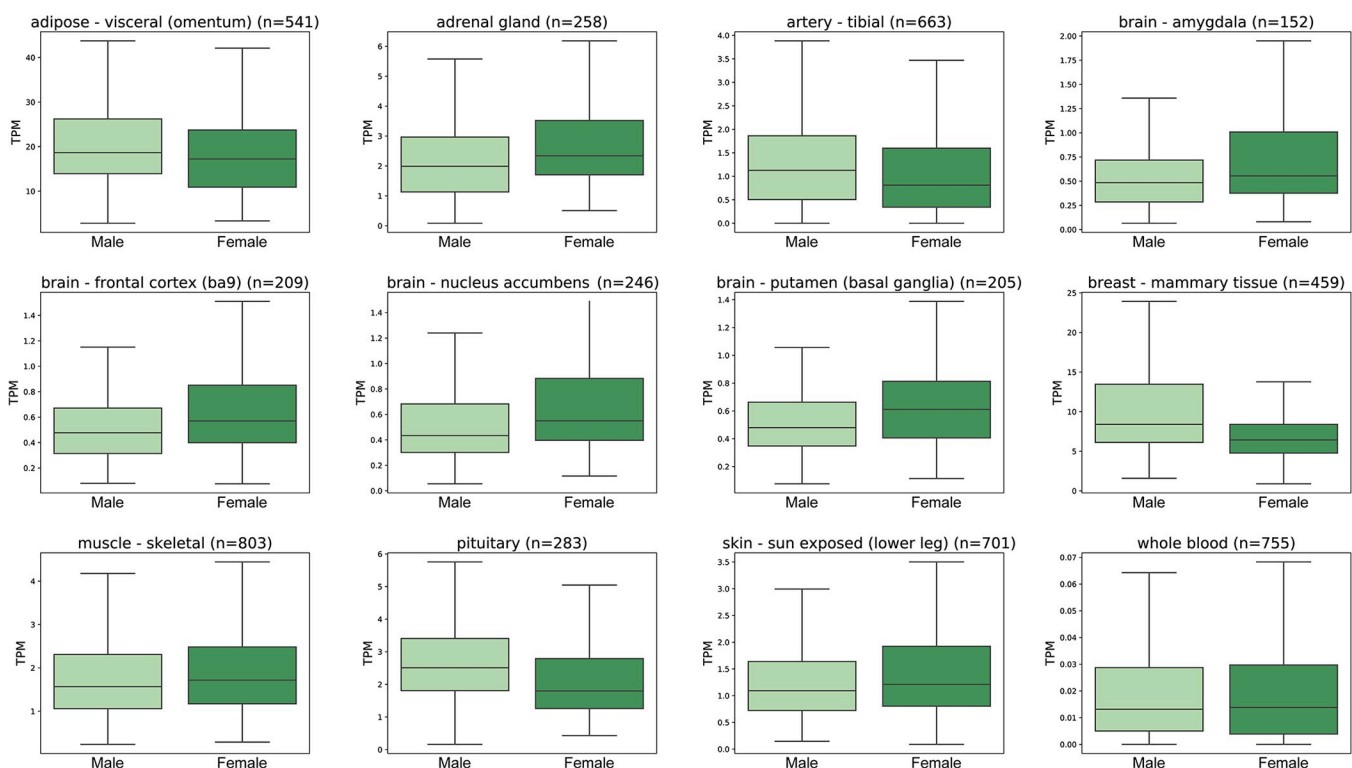

**Fig 9. ENPEP mRNA expression measured in TPM between the sexes in non-sex-related tissues.** The 11 tissues that were significantly different between the sexes (ANOVA; p-value ≤0.05). Each boxplot displays the distribution of expression values for the gene-tissue intersection: the box represents the interquartile range (IQR), the lower boundary marks the 1st quartile (Q1), the upper boundary marks the 3rd quartile (Q3), and the horizontal line within the box indicates the median. The whiskers extend to the maximum and minimum values within 1.5 times the IQR. Data derived from the GTEx dataset (v8) (https://www.gtexportal.org/home/) [45].

**Table 3. Genes strongly correlated with ENPEP expression in various tissues.**

| Tissue | Correlated gene | Correlation | HGNC | UniProt | Description | Panther protein class |
|---|---|---|---|---|---|---|
| All tissues | NRP1 | 0.8015 | 8004 | O14786 | Neuropilin 1 | |
| All tissues | ADGRL4 | 0.7764 | 20822 | Q9HBW9 | Adhesion G protein-coupled receptor L4 | G-protein coupled receptor |
| All tissues | TFPI | 0.7649 | 11760 | P10646 | Tissue factor pathway inhibitor | protease inhibitor |
| All tissues | VEGFC | 0.7628 | 12682 | P49767 | Vascular endothelial growth factor C | growth factor |
| All tissues | GNG11 | 0.7473 | 4403 | P61952 | Guanine nucleotide-binding protein G(I)/G(S)/G(O) subunit gamma-11 | heterotrimeric G-protein |
| All tissues | ROBO4 | 0.7409 | 17985 | Q8WZ75 | Roundabout homolog 4 | scaffold/adaptor protein |
| All tissues | IGFBP4 | 0.7351 | 5473 | P22692 | Insulin-like growth factor-binding protein 4 | protease inhibitor |
| All tissues | CFI | 0.7267 | 5394 | P05156 | Complement factor I | serine protease |
| All tissues | ENG | 0.7238 | 3349 | P17813 | Endoglin | transmembrane signal receptor |
| All tissues | CXCL12 | 0.7173 | 10672 | P48061 | Stromal cell-derived factor 1 | |
| Terminal ileum | SLC3A1 | 0.9682 | 11025 | Q07837 | Neutral and basic amino acid transport protein rBAT | amylase |
| Terminal ileum | ACE2 | 0.9653 | 13557 | Q9BYF1 | Angiotensin-converting enzyme 2 | metalloprotease |
| Terminal ileum | MEP1A | 0.9651 | 7015 | Q16819 | Meprin A subunit alpha | metalloprotease |
| Terminal ileum | ANPEP | 0.9636 | 500 | P15144 | Aminopeptidase N | metalloprotease |
| Terminal ileum | MEP1B | 0.9631 | 7020 | Q16820 | Meprin A subunit beta | metalloprotease |
| Terminal ileum | PLA2G12B | 0.9611 | 18555 | Q9BX93 | Group XIIB secretory phospholipase A2-like protein | phospholipase |
| Terminal ileum | DPP4 | 0.9609 | 3009 | P27487 | Dipeptidyl peptidase 4 | serine protease |
| Terminal ileum | ALDOB | 0.9593 | 417 | P05062 | Fructose-bisphosphate aldolase B | aldolase |
| Terminal ileum | SLC5A1 | 0.9592 | 11036 | P13866 | Sodium/glucose cotransporter 1 | secondary carrier transporter |
| Terminal ileum | CES2 | 0.959 | 1864 | O00748 | Cocaine esterase | esterase |
| Kidney cortex | LRP2 | 0.8704 | 6694 | P98164 | Low-density lipoprotein receptor-related protein 2 | |
| Kidney cortex | GJB2 | 0.8528 | 4284 | P29033 | Gap junction beta-2 protein | gap junction |
| Kidney cortex | TINAG | 0.8463 | 14599 | Q9UJW2 | Tubulointerstitial nephritis antigen | cysteine protease |
| Kidney cortex | UGT2A3 | 0.8434 | 28528 | Q6UWM9 | UDP-glucuronosyltransferase 2A3 | |
| Kidney cortex | UGT1A9 | 0.8335 | 12541 | O60656 | UDP-glucuronosyltransferase 1A9 | glycosyltransferase |
| Kidney cortex | ANPEP | 0.8259 | 500 | P15144 | Aminopeptidase N | metalloprotease |
| Kidney cortex | SLC4A4 | 0.8179 | 11030 | Q9Y6R1 | Electrogenic sodium bicarbonate cotransporter 1 | secondary carrier transporter |
| Kidney cortex | TCN2 | 0.8102 | 11653 | P20062 | Transcobalamin-2 | |
| Kidney cortex | ENTPD5 | 0.8081 | 3367 | O75356 | Ectonucleoside triphosphate diphosphohydrolase 5 | nucleotide phosphatase |
| Kidney cortex | SLC17A1 | 0.8066 | 10929 | Q14916 | Sodium-dependent phosphate transport protein 1 | secondary carrier transporter |
| Kidney cortex | ACE2 | 0.8062 | 13557 | Q9BYF1 | Angiotensin-converting enzyme 2 | metalloprotease |
| Testis | STOM | 0.7812 | 3383 | P27105 | Erythrocyte band 7 integral membrane protein | cytoskeletal protein |
| Testis | ETS1 | 0.7682 | 3488 | P14921 | Protein C-ets-1 | winged helix/forkhead transcription factor |
| Testis | PDLIM5 | 0.7569 | 17468 | Q96HC4 | PDZ and LIM domain protein 5 | actin or actin-binding cytoskeletal protein |
| Testis | IFI16 | 0.7557 | 5395 | Q16666 | Gamma-interferon-inducible protein 16 | DNA-binding transcription factor |
| Testis | A2M | 0.7527 | 7 | P01023 | Alpha-2-macroglobulin | protease inhibitor |
| Testis | ZFP36L2 | 0.7503 | 1108 | P47974 | mRNA decay activator protein ZFP36L2 | RNA metabolism protein |

*(Continued)*

**Table 3.** (Continued)

| Tissue | Correlated gene | Correlation | HGNC | UniProt | Description | Panther protein class |
|---|---|---|---|---|---|---|
| **Testis** | KCNJ8 | 0.7488 | 6269 | Q15842 | ATP-sensitive inward rectifier potassium channel 8 | ion channel |
| **Testis** | NRP1 | 0.7483 | 8004 | O14786 | Neuropilin-1 | |
| **Testis** | EDNRB | 0.747 | 3180 | P24530 | Endothelin receptor type B | |
| **Testis** | OSMR | 0.7455 | 8507 | Q99650 | Oncostatin-M-specific receptor subunit beta | transmembrane signal receptor |
| **Transverse colon** | EMCN | 0.7412 | 16041 | Q9ULC0 | Extracellular matrix protein 2 | |
| **Transverse colon** | ADGRL4 | 0.7313 | 20822 | Q9HBW9 | Adhesion G protein-coupled receptor L4 | G-protein coupled receptor |
| **Transverse colon** | FAM198B | 0.731 | 25312 | Q6UWH4 | Golgi-associated kinase 1B | |
| **Transverse colon** | PREX2 | 0.7299 | 22950 | Q70Z35 | Phosphatidylinositol 3,4,5-trisphosphate-dependent Rac exchanger 2 protein | guanyl-nucleotide exchange factor |
| **Transverse colon** | NRP1 | 0.7214 | 8004 | O14786 | Neuropilin-1 | |
| **Liver** | GOLIM4 | 0.7913 | 15448 | O00461 | Golgi integral membrane protein 4 | |
| **Liver** | TM9SF2 | 0.7891 | 11865 | Q99805 | Transmembrane 9 superfamily member 2 | transporter |
| **Liver** | ITFG1 | 0.7878 | 30697 | Q8TB96 | T-cell immunomodulatory protein | |
| **Liver** | TMEM59 | 0.7876 | 1239 | Q9BXS4 | Transmembrane protein 59 | |
| **Liver** | GC | 0.7768 | 4187 | P02774 | Vitamin D-binding protein | transfer/carrier protein |
| **Visceral adipose** | MYO1B | 0.6423 | 7596 | O43795 | Unconventional myosin-Ib | actin binding motor protein |
| **Visceral adipose** | EMCN | 0.6083 | 16041 | Q9ULC0 | Endomucin | |
| **Visceral adipose** | NRP1 | 0.5907 | 8004 | O14786 | Neuropilin-1 | |
| **Visceral adipose** | TCF4 | 0.5875 | 11634 | P15884 | Transcription factor 4 | basic helix-loop-helix transcription factor |
| **Visceral adipose** | GUCY1A2 | 0.5727 | 4684 | P33402 | Guanylate cyclase soluble subunit alpha-2 | guanylate cyclase |
| **Lung** | SLIT3 | 0.6335 | 11087 | O75094 | Slit homolog 3 protein | |
| **Lung** | EDNRA | 0.5654 | 3179 | P25101 | Endothelin-1 receptor | |
| **Lung** | ANTXR1 | 0.5609 | 21014 | Q9H6X2 | Anthrax toxin receptor 1 | cell adhesion molecule |
| **Lung** | ECM2 | 0.5604 | 3154 | O94769 | Extracellular matrix protein 2 | |
| **Lung** | MYH10 | 0.5541 | 7568 | P35580 | Myosin-10 | |

ENPEP (>0.95), most of which were associated with enterocyte function (S9 Table). Zinc metalloenzymes akin to ENPEP, such as ACE2, MEP1A, ANPEP and MEP1B, were among the five most strongly correlated genes, as shown in Table 3. However, this very strong correlation between ENPEP and ACE2 was not detected in other parts of the gastrointestinal tract, such as the esophagus, stomach, or colon (S9 Table). A very strong correlation of ENPEP (>0.8) with ANPEP and ACE2 was also detected in the kidney cortex (Table 3). Conversely, overall correlations were weaker in the lung, with a maximum correlation of 0.63 (Table 3).

When comparing ENPEP and several known host factors for SARS-CoV-2 (ACE2, DPP4, TMPRSS2, NRP1, and CTSL), and ANPEP, a zinc metalloenzyme similar to ENPEP and known human coronavirus receptor, the correlations of expression between them varied greatly from tissue to tissue (S1 Fig). The small intestine is one of the most polarized tissues, with only strongly positive or negative correlations. NRP1, the gene most strongly correlated with ENPEP in all tissue samples, was positively correlated in all tissue samples except the

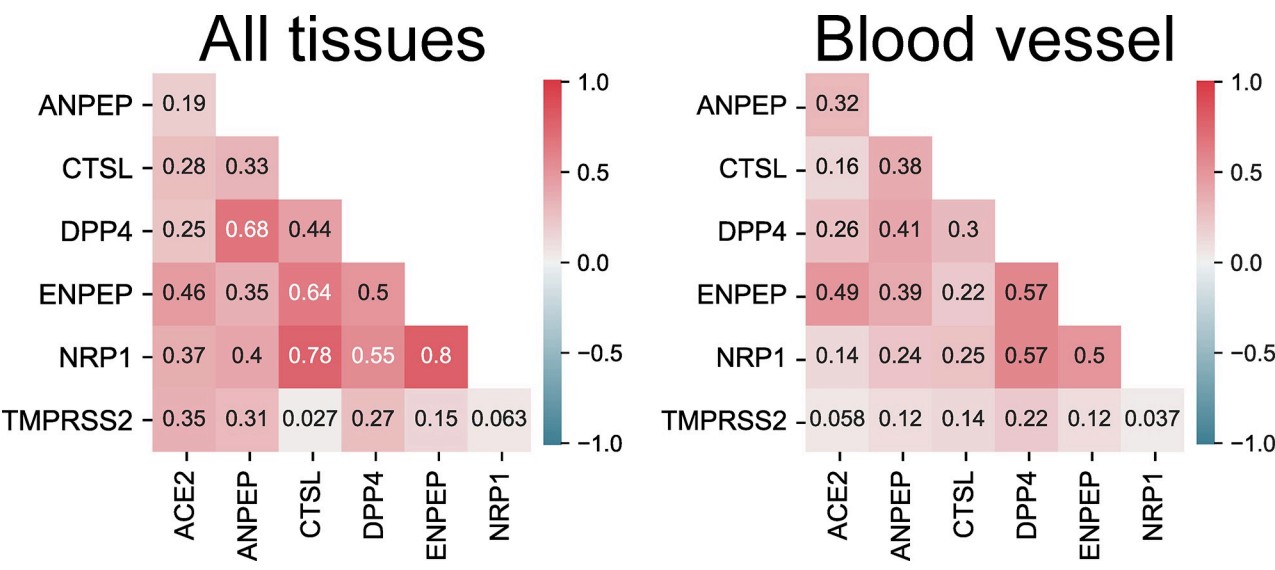

**Fig 10. A matrix plot of the correlations between ENPEP and other genes of interest in all tissues and blood vessels.** ACE2, angiotensin converting enzyme 2; ANPEP, alanyl aminopeptidase; CTSL, cathepsin L; DPP4, dipeptidyl peptidase 4; ENPEP, glutamyl aminopeptidase; NRP1, neuropilin 1; TMPRSS2, transmembrane serine protease 2. Data derived from the GTEx dataset (https://www.gtexportal.org/home/) [45].

small intestine, with a strong positive correlation ($\geq$0.7) in several tissues, such as the adrenal gland, breast, colon, esophagus, stomach, skin, testis, and vagina (S1 Fig).

Analyses performed using all samples from all cataloged tissues ('all tissues') revealed that all of the above mentioned genes had an overall positive correlation with ACE2, but ENPEP had the strongest correlation with ACE2 (0.46) by a small margin (Fig 10). This effect was more pronounced in blood vessels, where the correlation between ENPEP and ACE2 was moderate (0.49), compared to the runner-up ANPEP, which exhibited a weak correlation with ACE2 (0.32) (Fig 10). TMPRSS2, the classic cofactor responsible for priming the viral S protein, had a very weak correlation with ACE2 in blood vessels (0.058) (Fig 10).

### ENPEP strongly associated with vascular development in GO enrichment analysis

Gene ontology enrichment analysis was used to study the biological processes most associated with the genes coexpressed with ENPEP, identifying processes strongly correlated with ENPEP in select tissues. The most statistically significant processes in each of these tissues are presented in Table 4. Using expression values from all tissues in the GTEx database and a high correlation cutoff of 0.7 [49], 49 statistically significant biological process ontologies were found. Most of these were related to the function and development of the circulatory system, with the five most significant ontologies being 'blood vessel morphogenesis [GO:0048514]', 'blood vessel development [GO:0001568]', 'vasculature development [GO:0001944]', 'cardiovascular system development [GO:0072358]', and 'circulatory system development [GO:0072359]'. Terms related to endothelial function and chemotaxis were also identified.

The gene ontologies associated with ENPEP were also investigated using artery, intestine, kidney, liver, testis, whole blood, skeletal muscle, and lung data. Tibial artery samples associated ENPEP with many of the same vascular processes, such as 'vasculature development [GO:0001944]' and 'angiogenesis [GO:0001525]', as observed in comparison of all tissues. Different ontologies were present in the kidney, mostly related to the transport of various small

**Table 4. Gene ontology enrichment analysis of genes strongly correlated with ENPEP.**

| source | Tissue | native | Process | p-value |
|---|---|---|---|---|
| GO:BP | All tissues | GO:0048514 | blood vessel morphogenesis | 4.19E-09 |
| GO:BP | All tissues | GO:0001568 | blood vessel development | 1.67E-08 |
| GO:BP | All tissues | GO:0001944 | vasculature development | 2.82E-08 |
| GO:BP | All tissues | GO:0072358 | cardiovascular system development | 3.20E-08 |
| GO:BP | All tissues | GO:0072359 | circulatory system development | 1.06E-07 |
| GO:BP | All tissues | GO:0035239 | tube morphogenesis | 1.45E-07 |
| GO:BP | All tissues | GO:0001525 | angiogenesis | 7.60E-07 |
| GO:BP | All tissues | GO:0035295 | tube development | 1.33E-06 |
| GO:BP | All tissues | GO:0045766 | positive regulation of angiogenesis | 0.000186 |
| GO:BP | All tissues | GO:0001938 | positive regulation of endothelial cell proliferation | 0.000307 |
| GO:BP | All tissues | GO:0045765 | regulation of angiogenesis | 0.00037 |
| GO:BP | All tissues | GO:1904018 | positive regulation of vasculature development | 0.000414 |
| GO:BP | Artery—tibial | GO:0001944 | vasculature development | 7.73E-09 |
| GO:BP | Artery—tibial | GO:0072358 | cardiovascular system development | 9.66E-09 |
| GO:BP | Artery—tibial | GO:0001568 | blood vessel development | 2.50E-08 |
| GO:BP | Artery—tibial | GO:0001525 | angiogenesis | 8.99E-07 |
| GO:BP | Artery—tibial | GO:0048514 | blood vessel morphogenesis | 1.32E-06 |
| GO:BP | Small intestine—terminal ileum | GO:0006629 | lipid metabolic process | 1.11E-34 |
| GO:BP | Small intestine—terminal ileum | GO:0044281 | small molecule metabolic process | 1.70E-34 |
| GO:BP | Small intestine—terminal ileum | GO:0006082 | organic acid metabolic process | 7.63E-27 |
| GO:BP | Small intestine—terminal ileum | GO:0043436 | oxoacid metabolic process | 2.49E-25 |
| GO:BP | Small intestine—terminal ileum | GO:0044255 | cellular lipid metabolic process | 1.85E-24 |
| GO:BP | Colon—transverse | GO:0001568 | blood vessel development | 1.53E-10 |
| GO:BP | Colon—transverse | GO:0035239 | tube morphogenesis | 3.38E-10 |
| GO:BP | Colon—transverse | GO:0001944 | vasculature development | 3.42E-10 |
| GO:BP | Colon—transverse | GO:0072358 | cardiovascular system development | 4.16E-10 |
| GO:BP | Colon—transverse | GO:0035295 | tube development | 1.07E-09 |
| GO:BP | Kidney—cortex | GO:0015711 | organic anion transport | 8.97E-08 |
| GO:BP | Kidney—cortex | GO:0006820 | anion transport | 3.61E-06 |
| GO:BP | Kidney—cortex | GO:0006855 | drug transmembrane transport | 0.000781 |
| GO:BP | Kidney—cortex | GO:0046942 | carboxylic acid transport | 0.001102 |
| GO:BP | Kidney—cortex | GO:0015849 | organic acid transport | 0.001163 |
| GO:BP | Testis | GO:0032501 | multicellular organismal process | 2.07E-06 |
| GO:BP | Testis | GO:0010033 | response to organic substance | 2.43E-06 |
| GO:BP | Testis | GO:0032502 | developmental process | 6.85E-06 |
| GO:BP | Testis | GO:0048856 | anatomical structure development | 3.26E-05 |
| GO:BP | Testis | GO:0009653 | anatomical structure morphogenesis | 4.47E-05 |
| GO:BP | Liver | GO:0030449 | regulation of complement activation | 2.63E-06 |
| GO:BP | Liver | GO:0002920 | regulation of humoral immune response | 1.25E-05 |
| GO:BP | Liver | GO:0006956 | complement activation | 0.000269 |
| GO:BP | Liver | GO:0002252 | immune effector process | 0.000334 |
| GO:BP | Liver | GO:0002576 | platelet degranulation | 0.002001 |

molecules. In the liver, the most significant terms were 'regulation of complement activation [GO:0030449]', 'regulation of humoral immune response [GO:0072358]' and 'complement activation [GO:0001568]', which link ENPEP to the complement system. Testis data associated ENPEP with terms such as 'developmental process [GO:0032502]', 'anatomical structure development [GO:0048856]', and 'anatomical structure morphogenesis [GO:0009653]'.

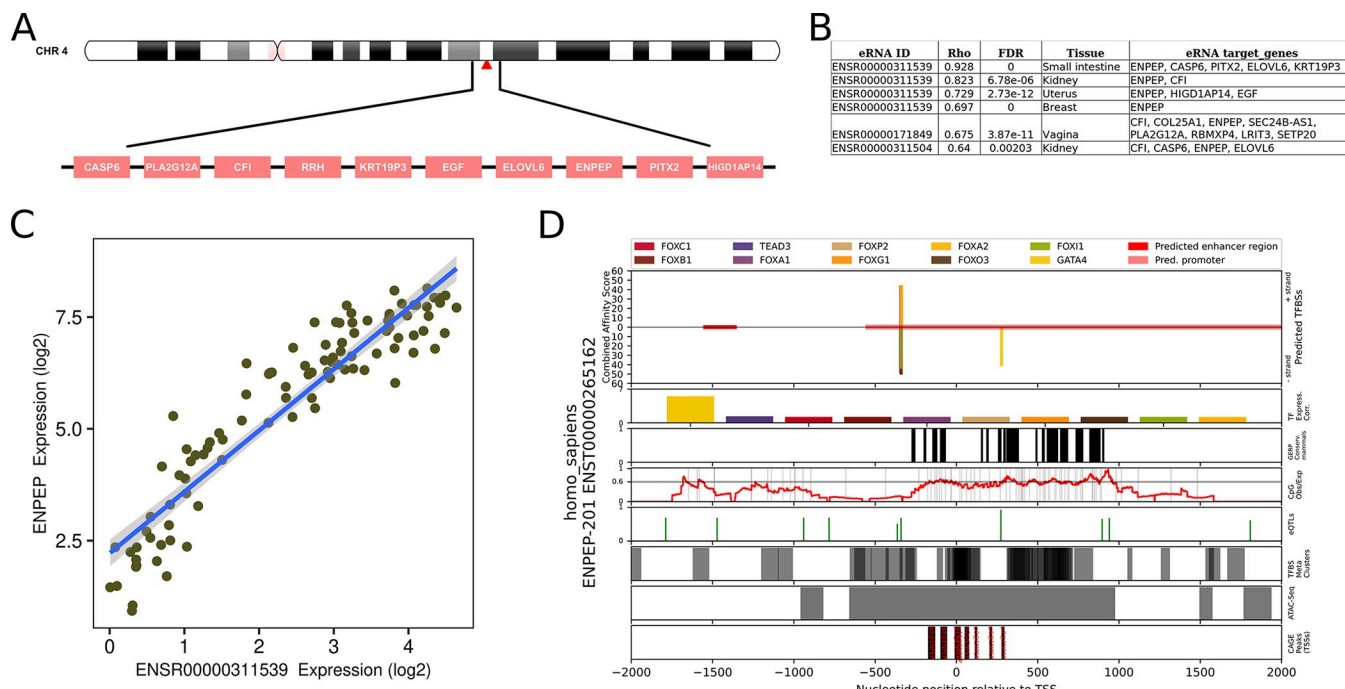

**Fig 11. Genomic and regulatory context of the ENPEP gene.** (A) The ENPEP gene is located on chromosome 4 in proximity to several genes that are co-regulated in several tissues (small intestine, kidney, uterus, and breast) by the same enhancer (ENSR00000311539), as identified in the HeRA database (B). The expression of the identified HeRA enhancer (ENSR00000311539) is strongly correlated with that of ENPEP (>0.9) in the small intestine (C). Transcription factor binding site prediction of the ENPEP promoter was performed on the region +/- 2,000 base pairs relative to the TSS of the primary protein-coding ENPEP transcript (ENST00000265162) (D). This analysis revealed a FOX family binding site cluster at -349 to -332 base pairs relative to the TSS (upstream) and a GATA4 binding site at 272 to 283 base pairs downstream of the TSS.

The small intestine, a major locus of ENPEP expression, revealed hundreds of significant biological process ontologies, with the top 10 most significant being various metabolic processes. Conversely, analysis of the colon revealed processes primarily involved in angiogenesis, implying that ENPEP is not as involved in the digestive functions of the colon. The lung, skeletal muscle and whole blood did not produce any associated ontologies (Table 4).

## ENPEP transcription factors, genomic context, and enhancers

TFBS prediction was performed on the promoter of the single protein-coding transcript (ENST00000265162) of the ENPEP gene. The results showed that a FOX family TF cluster is present ~350 bp upstream of the TSS of the ENPEP transcript (Fig 11). The cluster overlaps a variant (rs4458518) cataloged in the [45] known to upregulate the expression of ENPEP in the lung (Fig 11). This result aligns with the results of enrichment analysis of genes coexpressed with ENPEP (≥0.50; all tissues) in the GTEx dataset using the MSigDB (vers. 2023.2) [51] gene sets for TF target genes, where STAT3 (odds value, 21.3; BH p-value, 0.028), STAT5B (odds value, 15.48; BH p-value, $2.57 \times 10^{-11}$), STAT5A (odds value, 15.28; BH p-value, $2.57 \times 10^{-11}$), and FOXD3 (odds value, 14.20; BH p-value, 0.043) target genes showed the strongest level of enrichment (S10 Table).

## Discussion

Comparing our results with our previous study on the distribution and characteristics of ACE2, it appears that ENPEP and ACE2 are involved in very similar biological processes,

sharing 4 out of the top 5 biological process ontologies: blood vessel morphogenesis, blood vessel development, vasculature development and cardiovascular system development [9]. Notably, angiogenesis is the process most strongly correlated with ENPEP rather than its classically known role in the degradation of ANG2 in the RAAS. Furthermore, NRP1, the gene with the highest overall correlation to ENPEP, is also intimately involved in vascular development, functioning as a receptor for angiogenic peptides such as VEGF165 and SEMA3A and possibly promoting blood vessel permeability [65]. Other genes strongly related to angiogenesis, such as ADGRL4 (adhesion G protein-coupled receptor L4) [66], TFPI (tissue factor pathway inhibitor) [67], and VEGFC (vascular endothelial growth factor) [68] also ranked among the most correlated genes, further linking ENPEP to angiogenesis. The angiogenic properties of APA have been studied in vitro. Marchiò et al. demonstrated that human dermal microvascular endothelial cells exhibited decreased cell migration and inhibited cell proliferation when subjected to an APA inhibitor and decreased the formation of capillary-like structures when endothelial cells were grown on a synthetic basement membrane when APA activity was impaired [38]. APA also appears to have a role in the angiogenic response to hypoxia. Kubota et al. cultured endothelial cells from APA-knockout mice, resulting in decreased in vitro vascular formation when subjected to a hypoxic challenge [69]. This was further linked to decreased levels of hypoxia-inducible factor 1-alpha (HIF1a), suggesting that APA contributes to the stability of this central regulator of the hypoxic response [69, 70]. While these studies clearly demonstrate the angiogenic properties of APA, particularly under stresses such as hypoxia, the gene appears not to be strictly essential as demonstrated by the viability of APA-knockout mice [69]. APA upregulation in disease likely contributes to the regeneration of damaged vasculature, but the effects are not necessarily positive for the organism. Murine studies investigating the role of APA in post myocardial infarction have demonstrated RAAS dysregulation as a result of increased APA activity [71]. As further evidence, treatment of mice with a centrally-acting APA inhibitor (Firibastat) after myocardial infarction resulted in better preserved cardiac function compared to controls [72]. There is little, if any, research conducted on the possible interactions between APA and some of the highly correlated angiogenic proteins such as NRP1 and VEGFC. Inquiry into these relationships might uncover new mechanisms underlying angiogenesis.

Strong connections to angiogenesis and other processes related to vascular development imply roles for ENPEP in pathological states such as cancer, cardiovascular disease, and tissue remodeling. Fittingly, ENPEP has been connected to a variety of cancers and is upregulated in colorectal and renal neoplasms [41, 42]. Prior research also suggests that the expression of ENPEP is upregulated in other pathologic processes. Schlingemann et al. used immunostaining to demonstrate that APA is primarily present in tissues undergoing vascular generation, with tumor samples and damaged tissues such as inflamed synovial and granulation tissue exhibiting much more prominent staining than healthy vascular tissue, implying upregulation in tissues undergoing angiogenesis [73]. However, this upregulation is not universal. As an example, iron-deficient liver tumors which have been found to exhibit reduced APA levels [43], likely reflecting the diversity of conditions like cancer.

The many angiogenic properties linked to APA suggest a subset of cells closely associated with the vasculature where ENPEP is actively involved in these processes. Indeed, prior research has identified APA in the vasculature of the brain and periphery, with pericytes being implicated as the main vascular cell type that expresses ENPEP [38, 74]. Pericytes are specialized regulatory mural cells closely related to vascular smooth muscle cells [75, 76] that participate in guiding local angiogenesis, adjusting the permeability of the endothelium, and maintaining the microvasculature [77]. ENPEP has also been used as a gene marker for angiogenic pericytes in the intestine [59]. Our study revealed a similar expression pattern, clearly

observed in the human heart, where pericytes and smooth muscle cells were the cell populations with the most prevalent ENPEP expression. In the Human Lung Cell Atlas dataset, the most prevalent expression was observed in alveolar fibroblasts, which are cells located directly in the basal lamina between epithelial pneumocytes and the capillary endothelium, and electron microscopy revealed cytoplasmic processes wrapping around the capillaries [78]. These characteristics attributed to alveolar fibroblasts are remarkably similar to those of pericytes, defined by their localization in the basal lamina directly beneath the endothelium and a characteristic morphology consisting of cytoplasmic processes [77], suggesting that these alveolar fibroblasts might play similar roles in the lung parenchyma, guiding vessel development around the alveoli, with APA playing a regulatory role. Importantly, as we and others have shown previously, expression of ACE2 in the lung is very limited, with arterial vascular endothelial cells of the lung showing the greatest proportion of ACE2 expression at just 2.55% of cells [9, 79]. Our current analysis of scRNA-Seq data from the Human Protein Atlas data likewise shows limited expression of ACE2 in the lung, where the highest distribution of expression is observed in a type 2 alveolar cell cluster, with just 3.6% of cells expressing. By comparison, in the same dataset, ENPEP is expressed in 6.5% of lung fibroblasts and 6.5% of lung smooth muscle cells. This low level of expression of ACE2 in the primary organ of pathology in COVID-19 infection points to other functional causes of disease severity to which ENPEP may be a contributor, namely the RAS/RAAS.

The high expression of ENPEP in perivascular smooth muscle cells is supported by data from the Human Protein Atlas single-cell dataset. Indeed, it appears that specific populations of smooth muscle cells, most likely pericytes due to the expression of marker genes such as NOTCH3, PDGFRB, and STEAP4 [56, 80], express ENPEP throughout the body, with some vascular smooth muscle cell populations showing an over 50% prevalence of ENPEP expression. Adipose tissue well illustrates the vascular staining pattern of the APA protein: clear signals in both capillaries and arterioles, with no staining in the adipocytes themselves. Single-cell analysis of adipose tissue also revealed smooth muscle cells and fibroblasts as the main cell types with ENPEP expression rather than the adipose cells themselves. Despite vascular staining of APA, single-cell data reveals ENPEP to be almost completely absent from endothelial cells, indicating that the staining is in the mural population of cells, consisting of vascular smooth muscle cells and pericytes.

Furthermore, the vascular expression of ENPEP is likely upregulated in disease. Schlingemann et al. used immunostaining to demonstrate that APA is primarily present in tissues undergoing vascular generation, with tumor samples and damaged tissues such as inflamed synovial and granulation tissue exhibiting much more prominent staining than healthy vascular tissue, suggesting its upregulation in the pericytes of tissues undergoing angiogenesis [73]. Fittingly, ENPEP has been connected to a variety of cancers and is upregulated in colorectal and renal neoplasms [41, 42]. However, upregulation is not always the case, as iron-deficient liver tumors were found to exhibit reduced APA levels [43]. These regulatory changes would likely be reflected in single-cell data, but further study of ENPEP expression in disease-specific datasets is needed.

The vascular expression of ENPEP and ACE2—particularly in pericytes—might offer the most interesting area of overlap between the two genes. Both APA and ACE2 are present in pericytes [81], and the importance of these cells in COVID-19 has been highlighted by multiple studies. Pericytes possibly participate in the vascular inflammation and hypercoagulopathy [81, 82], myocarditis [83], and CNS symptoms observed in patients with COVID-19 [84, 85]. The lack of ACE2 expression in endothelial cells, except possibly in newly formed microvessels [86], further reinforces the idea that pericytes are the central vascular target of SARS-CoV-2 [87]. This might be particularly relevant in the central nervous system, where disruption of the

blood–brain barrier may contribute to the persistent neurological symptoms sometimes reported after SARS-CoV-2 infection [88]. As a cell type that influences capillary permeability, pericytes are known to play an important role in maintaining blood–brain barrier integrity [89]. While It appears that the expression profiles of ACE2 and ENPEP align in vasculature and particularly pericytes, we found that many of the known SARS-CoV-2 cofactors had weak correlations with ACE2 in blood vessels. TMPRSS2 in particular had a very weak correlation with ACE2 (0.058), whereas ENPEP had the strongest correlation (0.49) of the genes we tested. If SARS-CoV-2 interacts with APA it could help explain the vascular effects of COVID-19 since pericytes are a systemic reservoir of cells in which APA and ACE2 are more strongly correlated than many of the currently known cofactors. This pericyte-targeted infection could promote the systemic vascular inflammation, capillary leakage, and blood-brain-barrier disruption discussed above.

Dysregulation of the RAS/RAAS has been proposed to be a contributing factor to the pathophysiology of COVID-19 [36], with evidence indicating that elevated levels of ANG2 are positively correlated with increased disease severity [90–92]. Correspondingly, the use of ACE inhibitors (ACEIs) and angiotensin receptor blockers (ARBs) often produces better outcomes in patients with COVID-19 infection [93] and results in a lower incidence of flu [94], contrary to initial concerns regarding the resultant increase in the expression of ACE2 providing additional viral targets [95]. Importantly, ANG2 levels are also linked to severity in several other diseases, such as H7N9 [96], and in a swine model the induction of ANG2 alone was shown to produce many of the most detrimental effects of COVID-19 infection (including increased pulmonary artery pressure, reduced blood oxygenation, increased coagulation, disturbed lung perfusion, diffuse alveolar damage, and acute tubular necrosis) [36].

As it does in myocardial infarction [71], ENPEP could also negatively impact the balance of the RAAS during COVID-19, regardless of its potential as a coreceptor. While SARS-CoV-2 interferes with ACE2 production of the vasoprotective and anti-inflammatory peptide ANG1-7 from ANG2 [97], ENPEP is upregulated by inflammation [73]. ANG2 can then be preferentially cleaved to ANG3 by APA, which can result in the production of inflammatory cytokines such as interleukin 6 (IL-6) by binding AT1 receptors [98]—an effect likely pronounced in the CNS, where proinflammatory AT1 is the preferred receptor [99, 100]. A study of NSAID treatment in mice has shown that ACE2 is strongly downregulated and TMPRSS2 upregulated in the lung after NSAID treatment [101] with similar changes observed in rat kidney [102], supporting the idea that inflammation-associated changes could drive dysregulation of RAAS and impact cell infection rates. Many studies have investigated the use of angiotensin receptor blockers (ARBs) for the treatment of RAAS dysregulation in COVID-19, but no conclusive evidence for the efficacy of these treatments has emerged [103]. It will be interesting to see whether the novel antihypertensive APA-inhibitor firibastat [35] would counteract RAAS dysregulation in COVID-19 if the drug clears clinical trials.

The CNS role of ENPEP deserves special consideration, as APA appears to be one of the central enzymes in the brain renin-angiotensin axis, upregulating systemic blood pressure by producing ANGIII, which increases vasopressin release and sympathetic activity. It also modulates the baroreceptor reflex—an effect of great recent interest due to clinical trials of firibastat, a novel antihypertensive agent that functions as a centrally acting APA inhibitor [35]. A detailed study of the CNS distribution of APA in the human brain by de Mota et al. [74] revealed the greatest APA activity in highly vascular portions of the brain, such as the choroid plexus, pineal gland, and paraventricular areas of the hypothalamus, whereas brain parenchymal APA activity was found to be much more location dependent and was only present in certain areas, such as the medulla oblongata, prefrontal cortex, and olfactory bulb. In addition to APA activity assays, immunohistochemistry has been used to localize APA to both blood

vessels and neurons in the medulla oblongata [74]. Similar findings have also been observed in rats, with APA staining in cerebral microvasculature and the blood–brain barrier [104, 105]. Our analysis of the Human Protein Atlas single-cell datasets revealed ENPEP expression only in select neuron populations, likely reflecting the highly localized pattern observed by de Mota et al., with large sections of the cerebrum devoid of major APA activity. This could also explain the absence of APA staining in the immunohistochemistry images of the Human Protein Atlas.

Despite prior research localizing APA activity in the choroid plexus and paraventricular areas, human ependymal expression has not been confirmed with immunostaining. Rat ependymal samples have been found to stain positive for APA [106], but the same is not observed in all animal models, with gerbil subfornical areas lacking APA immunostaining [107]. The highly prevalent expression of ENPEP in human ependymal cells found in our analysis of single-cell data from the Human Brain Cell Atlas has not been previously documented. Indeed, it could be possible that the ependyma might exhibit some of the most prevalent ENPEP expression in certain areas of the brain, with ependymal cells possibly contributing to the CNS regulation of blood pressure. Further study and immunohistochemistry of the brain ependyma are necessary to confirm these findings.

In addition to the role of ENPEP in vasculature, our gene ontology and correlation analyses suggest several tissue-specific functions for the gene, with the most statistically significant processes differing from angiogenesis in certain tissues. In the kidney cortex, where ENPEP is traditionally known for its role in angiotensin catabolism, the top correlated genes encode proteins important for kidney function, such as low-density lipoprotein receptor-related protein 2 (LRP2), gap junction beta-2 protein (GJB2), and tubulointerstitial nephritis antigen (TINAG), which are associated with the reabsorption of proteins, permeability of small molecules, and kidney development, respectively [108–110]. The role of ENPEP in the kidney has been studied in detail, and murine studies have estimated that APA accounts for more than 60% of ANGIII production [111]. APA has also been suggested to protect against glomerular damage [39] and regulate glomerular function [112].

Less studied in the small intestine, ENPEP is likely involved in the end-stage digestion of proteins given its function as an aminopeptidase, its cellular localization at the brush border, its associated metabolic processes, and its strong correlation with other peptidases involved in the digestion of proteins, such as ACE2 and APN [113, 114]. Amino acid transport is another possible role for APA in the small intestine. The most correlated gene in this tissue is SLC3A1, which encodes the amino acid transporter heavy chain SLC3A1, whereas ACE2—the second most correlated gene—has also been shown to be involved in amino acid transport. The ACE2 protein interacts with the sodium-dependent neutral amino acid transporter B0AT1 (SLC6A19) to aid protein absorption, with ACE2 knockout mice correspondingly exhibiting impaired amino acid absorption [115, 116]. Other highly correlated genes of the gut, such as the meprin zinc metalloproteinases MEP1a and MEP1b, regulate gut inflammatory responses by exerting anti- and proinflammatory effects, respectively [117, 118].

The intestinal connection between APA and lipid metabolism noted by the gene ontologies is probably not as straightforward, as lipases rather than aminopeptidases function as the main digestive enzymes [119]. However, saturated dietary fats have been noted to increase APA activity in the serum and testes of mice, and the same could be true in other tissues [120, 121]. Many membrane proteins, including APN, an aminopeptidase very similar to APA, have been suggested to aid in intestinal cholesterol absorption, suggesting that APA could also influence cholesterol uptake [122, 123].

The liver, where APA is present in a subset of hepatocytes, links ENPEP to complement activation and humoral immunity. On the other hand, ENPEP expression was almost absent in the leukocyte populations of the Human Protein Atlas. The leukocyte expression of ENPEP

could change depending on immunological activation or the developmental stage of the cells, but it is unlikely that ENPEP plays a major role in cellular immunity. Several murine studies have explored the role of APA in cellular immunity, with the protein also known as murine B-lymphocyte differentiation antigen BP-1/6C3 because it is present in developing murine B cells under the influence of interleukin 7 (IL-7) [124, 125]. However, this role of APA in murine immune development appears to be minor, with APA-deficient mice exhibiting normal development and leukocyte function [126]. The connection between ENPEP and the complement system is much less explored. ENPEP and human complement factor 1 genes are located close to each other at chromosome 4q25 and appear to both be regulated by a common enhancer, but the significance of this association is unclear [127]. Further study is required to investigate the potential role of APA in the complement system.

Although studies such as that of Schlingemann et al. [73] revealed increased APA levels in pathological tissues, the effects of age and sex have not been comprehensively explored. Despite a prior study reporting that serum APA activity tends to increase with age [128], our analysis shows that this is likely not the case in all tissues, with ENPEP expression variably increasing or decreasing depending on the tissue, possibly reflecting physiological changes associated with age. For example, in the prostate, where ENPEP expression tends to increase with age, the increase could be related to local angiogenesis related to prostate growth [129]. In skeletal muscle, increased APA activity could result from a variety of factors, including low-level inflammation associated with aging [130] or increased adiposity of skeletal muscle [131]. Reduced expression in the aorta and left ventricle, on the other hand, could result from increased fibrosis and decreased numbers of vascular smooth muscle cells [132]. While tissue samples did not demonstrate uniform changes in ENPEP expression with age, this is yet to be studied in microvasculature and specific cell populations of interest such as pericytes. Sex did not appear to play a significant role in ENPEP tissue expression in any of the analyses we performed making it unlikely that the gene has a marked influence on the differences in cardiovascular health or disease susceptibility between sexes. Even in the tissues with statistically significant differences in ENPEP expression between sexes, the differences in expression were slight, with marked overlap between males and females.

When considering APA as a possible cofactor for SARS-CoV-2, it is notable that human coronaviruses are known to use aminopeptidases such as APN (HCoV-229E) [17] and DPP4 (MERS-CoV) [18] to facilitate host cell invasion. APA is an aminopeptidase that was identified as a cofactor candidate by Qi et al. [8] and was identified as a possible viral receptor in two separate machine learning analyses of possible viral receptors [8, 20]. We discovered that of the other genes associated with SARS-CoV-2 (i.e., DPP4, TMPRSS2, NRP1, and CTSL), and ANPEP (encoding APN, the HCoV-229E receptor [17]), ENPEP has the strongest overall correlation with ACE2 when assessing its expression in all tissues (0.46) and in vasculature (0.49). However, despite the involvement of both ENPEP and ACE2 in angiogenesis and the RAAS, the genes are only variably correlated depending on the tissue, with gross mRNA expression between the two mainly aligning in the small intestine and, to a lesser extent, the kidney cortex. Although ENPEP and ACE2 correlate almost perfectly in the small intestine at 0.97 and both proteins localize in the enterocyte brush border, the same level of correlation is not observed elsewhere on a tissue-wide scale, with the kidney cortex exhibiting the second highest ENPEP-ACE2 correlation at 0.82. Immunostaining of the kidney cortex, the tissue with the second highest ENPEP-ACE2 correlation at 0.82, revealed overlap of both proteins only in proximal tubular cells, despite APA also being present in the glomeruli. The absence of ENPEP epithelial cell expression outside the small intestine also limits the areas where APA could encounter an outside pathogen, making it unlikely to have a role in the initial target cells of SARS-CoV-2: small intestine enterocytes appear to be the only surface cell type exhibiting significant ENPEP

expression. While SARS-CoV-2 does directly infect enterocytes [133], COVID-19 is neverthe-less primarily a respiratory illness primarily transmitted by droplets or aerosols [134]. While more study is required on the interplay of ACE2 and ENPEP in the intestine if APA is found to be a SARS-CoV-2 cofactor, the most interesting interactions between APA, ACE2, and SARS-CoV-2 are likely found in blood vessels, noting the overlapping expression profiles in pericytes [9, 73, 81] and the significance of the vascular system in COVID-19 [36, 82].

Research has uncovered many of the viral mechanisms for SARS-CoV-2, with TMPRSS2 acting as the main cofactor for the viral receptor ACE2, cleaving the viral spike protein result-ing in membrane fusion [135], NRP1 acting as a cofactor and possible secondary viral receptor in the absence of ACE2 [10, 136], and CTSL enabling endosomal SARS-CoV-2 entry into cells as an alternative to TMPRSS2 mediated membrane fusion [137]. DPP4, the MERS-CoV recep-tor, is also suggested to influence SARS-CoV-2 cell entry, but the evidence for this is less direct, mainly based on similarities between MERS-CoV and SARS-CoV-2 [138], several computa-tional binding studies between DPP4 and SARS-CoV-2 spike protein with differing results [139–141], and studies indicating DPP4 inhibitor drugs might have a positive effect on COVID-19 outcomes in certain patient groups [138, 142]. However, APA also being a zinc aminopeptidase like the known human coronavirus receptor APN [15, 27], ENPEP appears understudied as a coronavirus cofactor candidate, with no studies focusing specifically on pos-sible interactions between ENPEP and SARS-CoV-2.

In summary, both APA and ACE2 appear to be present in systemic vasculature and peri-cytes in particular—a cell type implicated to have a significant role in COVID-19 [81]. If SARS-CoV-2 utilizes APA as a cofactor to infect enterocytes, this could contribute to systemic inflammation in COVID-19 and influence other pericyte-mediated functions such as capillary and blood-brain-barrier permeability [77, 89]. ENPEP and ACE2 both have important roles in the RAAS, with RAAS dysregulation suggested as an important part of COVID-19 pathophysi-ology [36]. More detailed in vitro studies are required to prove whether APA directly interacts with SARS-CoV-2. However, if it functions as a viral cofactor, it could assist in the infection of systemic vasculature and contribute to the pathology seen in specific organs, such as the kid-ney [143], heart [83], and blood–brain barrier [144], as well as the chronic symptoms and inflammation observed in patients with long COVID-19 [145].

## Study limitations and future directions

The study performs several analyses based on RNA-Seq data, both bulk and single-cell. Actual levels of protein and RNA expression are often loosely correlated and thus high or low values of gene expression may not fully reflect the actual biological state of the tissues analyzed. Addi-tionally, scRNA-Seq has inherent limitations, including low sequencing depth and dropout events. Classification of cell types by gene signatures, especially for highly specialized cells or specific states, is an evolving field which leaves room for interpretation and future improve-ment. As this study provides an initial bioinformatics characterization, further experimental studies are merited which directly test the role of ENPEP as a potential co-factor in SARS-CoV-2 infection, and the roles of ENPEP and ACE2 in RAS/RAAS in COVID-19 pathophysi-ology. For the next investigatory steps, we envision co-immunoprecipitation investigation of ENPEP with the full SARS-CoV-2 proteome and RNA-Seq of ACE2-expressing airway epithe-lial cell lines in ENPEP knockdown, knockout, and overexpression.

## Supporting information

**S1 Fig. Tissue specific co-expression analysis of ENPEP and co-factor genes of interest in SARS-CoV-2 infection in 30 human tissues.** Bulk RNA-Seq expression data from the GTEx

dataset (v8) [45] was retrieved as TPM values and Spearman correlation analysis was performed using the SciPy [48] Python library and heatmaps for each tissue were generated using the Seaborn [47] and Matplotlib [46] Python libraries.
(TIF)

**S1 Table. Normalized ENPEP expression (nTPM) in smooth muscle cells and endothelial cells in all tissues from single-cell mRNA data from the Human Protein Atlas (https:// www.proteinatlas.org/) [64].** Each row includes the 50 most up and down-regulated differentially expressed genes (DEGs) for the corresponding cell type cluster.
(XLSX)

**S2 Table. Normalized ENPEP expression (nTPM) in all 20 annotated cell clusters in liver tissue from single-cell mRNA data from the Human Protein Atlas (https://www. proteinatlas.org/) [64].** Each row includes the 50 most up and down-regulated differentially expressed genes (DEGs) for the corresponding cell type cluster.
(XLSX)

**S3 Table. Normalized ENPEP expression (nTPM) in all annotated cell clusters in solid immune-related tissues (bone marrow, lymph node, spleen, and thymus) from single-cell mRNA data from the Human Protein Atlas (https://www.proteinatlas.org/) [64].** Each row includes the 50 most up and down-regulated differentially expressed genes (DEGs) for the corresponding cell type cluster.
(XLSX)

**S4 Table. Normalized ENPEP expression (nTPM) in all annotated cell clusters in lower airway tissues (bronchus and lung) from single-cell mRNA data from the Human Protein Atlas (https://www.proteinatlas.org/) [64].** Each row includes the 50 most up and down-regulated differentially expressed genes (DEGs) for the corresponding cell type cluster.
(XLSX)

**S5 Table. Normalized ENPEP expression (nTPM) in all annotated cell clusters in brain tissue from single-cell mRNA data from the Human Protein Atlas (https://www.proteinatlas. org/) [64].** Each row includes the 50 most up and down-regulated differentially expressed genes (DEGs) for the corresponding cell type cluster.
(XLSX)

**S6 Table. Normalized ENPEP expression (nTPM) in all annotated cell clusters in gastrointestinal tissues (colon, esophagus, rectum, small intestine, stomach, and tongue) from single-cell mRNA data from the Human Protein Atlas (https://www.proteinatlas.org/) [64].** Each row includes the 50 most up and down-regulated differentially expressed genes (DEGs) for the corresponding cell type cluster.
(XLSX)

**S7 Table. Normalized ENPEP expression (nTPM) in all 20 annotated cell clusters in heart tissue from single-cell mRNA data from the Human Protein Atlas (https://www. proteinatlas.org/) [64].** Each row includes the 50 most up and down-regulated differentially expressed genes (DEGs) for the corresponding cell type cluster.
(XLSX)

**S8 Table. 20 genes had a high level of correlation ($\geq$0.70) with ENPEP in analysis of all samples from all tissues in the GTEx dataset.** Bulk RNA-Seq expression data from the GTEx dataset (v8) [45] was retrieved as TPM values and Spearman correlation analysis was

performed using the SciPy [48] Python library.
(XLSX)

**S9 Table. All genes with a moderate absolute level of correlation ($\geq$0.5) with ENPEP in analysis of all samples from gastrointestinal tissues in the GTEx dataset.** Bulk RNA-Seq expression data from the GTEx dataset (v8) [45] was retrieved as TPM values and Spearman correlation analysis was performed using the SciPy [48] Python library.
(XLSX)

**S10 Table. Transcription factor target enrichment analysis.** Fisher's exact test enrichment analysis of genes moderately coexpressed with ENPEP ($\geq$0.50) in all samples from all tissues in the GTEx dataset was performed using the transcription factor targets (TFT) gene set cataloged in the MSigDB (vers. 2023.2) database [51].
(XLSX)

# Author Contributions

**Conceptualization:** Antti Arppo, Harlan Barker, Seppo Parkkila.

**Data curation:** Harlan Barker.

**Formal analysis:** Harlan Barker.

**Funding acquisition:** Seppo Parkkila.

**Investigation:** Antti Arppo, Harlan Barker.

**Methodology:** Harlan Barker.

**Project administration:** Antti Arppo, Harlan Barker, Seppo Parkkila.

**Resources:** Harlan Barker, Seppo Parkkila.

**Software:** Harlan Barker.

**Supervision:** Seppo Parkkila.

**Validation:** Harlan Barker.

**Visualization:** Antti Arppo, Harlan Barker.

**Writing – original draft:** Antti Arppo, Harlan Barker, Seppo Parkkila.

**Writing – review & editing:** Antti Arppo, Harlan Barker, Seppo Parkkila.

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
