## [Decision Letter · Decision Letter 0]

3 Oct 2024

PONE-D-24-28420Bioinformatic characterization of ENPEP, the gene encoding a potential cofactor for SARS-CoV-2 infectionPLOS ONE

Dear Dr. Barker,

Thank you for submitting your manuscript to PLOS ONE. After careful consideration, we feel that it has merit but does not fully meet PLOS ONE’s publication criteria as it currently stands. Therefore, we invite you to submit a revised version of the manuscript that addresses the points raised during the review process.

We look forward to receiving your revised manuscript.

Kind regards,

Cheorl-Ho Kim, Ph.D.

Academic Editor

PLOS ONE

Journal Requirements:

Additional Editor Comments (if provided):

Dear Dr Barker

Thank you for your kind submission of your study and I am sorry for my delayed response due to difficulty to invite reviewers.

I have obtained a review with its positive outcome. I have also read due to the above reason and viewed its priority to consider.

I am satisfied with the review and would invite you to revise the versnio.

Thanks a lot

Sincerely

Cheorl-Ho Kim

Editors

Reviewers' comments:

Reviewer's Responses to Questions

**Comments to the Author**

1. Is the manuscript technically sound, and do the data support the conclusions?

Reviewer #1: Yes

2. Has the statistical analysis been performed appropriately and rigorously? 

Reviewer #1: Yes

3. Have the authors made all data underlying the findings in their manuscript fully available?

Reviewer #1: Yes

4. Is the manuscript presented in an intelligible fashion and written in standard English?

Reviewer #1: Yes

5. Review Comments to the Author

Reviewer #1: General Comments:

The manuscript provides a detailed bioinformatic analysis of ENPEP (encoding glutamyl aminopeptidase), focusing on its expression in various tissues and its potential role as a cofactor in SARS-CoV-2 infection. The use of large-scale publicly available datasets, including GTEx and single-cell RNA-Seq data, to assess the expression and function of ENPEP is commendable. The study explores correlations between ENPEP and ACE2, the primary receptor for SARS-CoV-2, and highlights the need for further experimental validation of ENPEP's role as a viral receptor. The manuscript is well-structured, with a clear flow of logic and detailed methodology.

However, there are a few areas that require clarification or improvement to strengthen the manuscript.

Specific Comments:

1. Rationale and Hypothesis:

While the manuscript clearly articulates the importance of studying ENPEP in the context of SARS-CoV-2 infection, it would benefit from a more explicit statement of the underlying hypothesis. The rationale behind choosing ENPEP as a potential cofactor for SARS-CoV-2 should be emphasized more, particularly in comparison to other cofactors like TMPRSS2 and NRP1. A discussion of why ENPEP, despite its relatively low expression in the lung, is of interest would add to the manuscript's depth.

2. Methodological Clarity:

The bioinformatic methods used in this study are comprehensive; however, the manuscript could benefit from a more detailed explanation of certain steps, especially for readers less familiar with bioinformatics. For instance, more clarity on the thresholds used for coexpression analysis and gene ontology enrichment would be helpful. It is also important to include justification for selecting specific datasets and cutoff values in the analyses.

3. Single-cell RNA-Seq Analysis:

The single-cell RNA-Seq data provides valuable insights, particularly regarding ENPEP expression in specific cell types like pericytes and enterocytes. However, it is recommended that the authors further elaborate on the potential functional implications of ENPEP expression in these cells, particularly in the context of viral entry and infection. Additionally, the authors should discuss the limitations of using scRNA-Seq data, such as potential dropout effects or the challenge of distinguishing between similar cell types.

4. Experimental Validation:

While the manuscript provides compelling bioinformatic evidence for ENPEP as a potential cofactor, it would greatly benefit from experimental validation, as suggested by the authors. The authors should discuss the feasibility of experiments, such as knockdown or overexpression studies, to validate ENPEP's role in SARS-CoV-2 infection. If experimental validation is beyond the scope of this study, the authors should clearly indicate this as a limitation and suggest future directions for experimental follow-up.

5. Figure Presentation:

The figures, particularly those presenting tissue-specific expression and immunohistochemistry images, are informative. However, some figures could be improved for clarity. For example, Figure 1 presents tissue-specific ENPEP expression, but it would be helpful to include additional labeling or annotations to make it more accessible to readers. The immunohistochemistry images should also be clearly labeled with magnification and staining details for better interpretation.

6. Age and Sex-related Differences:

The analysis of age and sex-related differences in ENPEP expression is an interesting aspect of the study. However, the biological significance of these findings should be further discussed. How do these differences relate to known age and sex-related differences in COVID-19 severity or susceptibility? Exploring these connections in greater detail would enhance the impact of the findings.

7. Discussion and Interpretation:

The discussion section could benefit from a more thorough comparison of the findings with existing literature on other known SARS-CoV-2 cofactors. Additionally, the implications of the strong correlation between ENPEP and angiogenesis-related genes like NRP1 should be explored further. How might ENPEP contribute to the vascular complications often observed in severe COVID-19 cases?

Conclusion:

Overall, this manuscript offers valuable insights into the potential role of ENPEP in SARS-CoV-2 infection. With minor revisions, particularly in the areas of methodological clarity and discussion of the biological significance of the findings, the manuscript has the potential to make a significant contribution to the field.

Recommendation:

I recommend this manuscript for publication with minor revisions.

I recommend to cite following in discussion:

1- J. Al-Awaida, W., Jawabrah Al Hourani, B., Swedan, S., Nimer, R., Alzoughool, F., J. Al-Ameer, H., ... & R. Hadi, N. (2021). Correlates of SARS-CoV-2 Variants on Deaths, Case Incidence and Case Fatality Ratio among the Continents for the Period of 1 December 2020 to 15 March 2021. Genes, 12(7), 1061.

2- Khirfan, F., Jarrar, Y., Al-Qirim, T., Goh, K. W., Jarrar, Q., Ardianto, C., ... & Ming, L. C. (2022). Analgesics induce alterations in the expression of SARS-CoV-2 entry and arachidonic-acid-metabolizing genes in the mouse lungs. Pharmaceuticals, 15(6), 696.

3- Hatmal, M. M. M., Al-Hatamleh, M. A., Olaimat, A. N., Mohamud, R., Fawaz, M., Kateeb, E. T., ... & Bindayna, K. M. (2022). Reported adverse effects and attitudes among Arab populations following COVID-19 vaccination: a large-scale multinational study implementing machine learning tools in predicting post-vaccination adverse effects based on predisposing factors. Vaccines, 10(3), 366.

6. PLOS authors have the option to publish the peer review history of their article (what does this mean?). If published, this will include your full peer review and any attached files.

Reviewer #1: No

---

## [Author Response · Author response to Decision Letter 0]

18 Nov 2024

REVIEWER COMMENTS AND RESPONSES

Reviewer 1 Comment 1

The manuscript provides a detailed bioinformatic analysis of ENPEP (encoding glutamyl aminopeptidase), focusing on its expression in various tissues and its potential role as a cofactor in SARS-CoV-2 infection. The use of large-scale publicly available datasets, including GTEx and single-cell RNA-Seq data, to assess the expression and function of ENPEP is commendable. The study explores correlations between ENPEP and ACE2, the primary receptor for SARS-CoV-2, and highlights the need for further experimental validation of ENPEP's role as a viral receptor. The manuscript is well-structured, with a clear flow of logic and detailed methodology.

However, there are a few areas that require clarification or improvement to strengthen the manuscript.

Author Response 1

We thank the reviewer for their appreciative and constructive comments on the manuscript. Below we address each point separately, and indicate all substantive changes made.

Reviewer 1 Comment 2

1. Rationale and Hypothesis:

While the manuscript clearly articulates the importance of studying ENPEP in the context of SARS-CoV-2 infection, it would benefit from a more explicit statement of the underlying hypothesis. The rationale behind choosing ENPEP as a potential cofactor for SARS-CoV-2 should be emphasized more, particularly in comparison to other cofactors like TMPRSS2 and NRP1.

Author Response 2

We thank the reviewer for this comment and we agree that additional statements on the central thesis of ENPEP being a gene and protein of potential interest in COVID-19 are merited. We have made the following additions to the Abstract, Introduction, and Discussion.

We have modified/added the following new text to the Abstract:

“Recent studies have proposed a role for ENPEP as a viral receptor in humans, and ENPEP and ACE2 are both closely involved in the renin-angiotensin-aldosterone system proposed to play an important role in SARS-CoV-2 pathophysiology.”

We have modified/added the following new text to the Introduction (italics indicates old text when new text is a modification of an existing sentence):

“The RAAS role of APA is of particular interest, as RAA dysregulation has been proposed to play an important part in SARS-CoV-2 pathophysiology, with impaired ACE2 function being linked to some of the deleterious cardiovascular effects seen in COVID-19 such as increased pulmonary artery pressure and coagulation in swine models (Rysz et al. 2021). APA might be a significant contributor to these adverse effects, due sharing ANG2 as a substrate with ACE2 (Te Riet et al. 2015). With impaired ACE2 function, ANG2 could instead be preferentially processed by APA.”

“Present in a broad range of tissues, including the small intestine, kidney cortex, liver, brain and vasculature, APA plays various general and tissue-specific roles, including activity in angiogenesis, kidney function, the endometrial cycle and implantation (L. Li et al. 1993; Marchiò et al. 2004; Velez et al. 2017; Mizutani et al. 2020), with corresponding connections to several disease processes, such as a variety of cancers, renal dysfunction, preeclampsia, and possibly COVID-19 via RAAS dysregulation (Blanco et al. 2014; Chuang et al. 2017; H. Wu et al. 2023; Velez et al. 2017; Hariyama et al. 2000).”

We have modified/added the following new text to the Discussion:

“Many studies have investigated the use of angiotensin receptor blockers (ARBs) for the treatment of RAAS dysregulation in COVID-19, but no conclusive evidence for the efficacy of these treatments has emerged (Matsuzawa et al. 2022). It will be interesting to see whether the novel antihypertensive APA-inhibitor firibastat (Khosla et al. 2022) would counteract RAAS dysregulation in COVID-19 if the drug clears clinical trials.”

Reviewer 1 Comment 3

A discussion of why ENPEP, despite its relatively low expression in the lung, is of interest would add to the manuscript's depth.

Author Response 3

This is indeed an interesting observation generally about both ENPEP and the known SARS-CoV-2 receptor ACE2 itself, both having low levels of expression in the lung. This may also be further evidence of a more complex situation underlying COVID-19 pathology, which ultimately involves dysregulation of RAS/RAAS and infection of circulatory cells and inflammation of vasculature.

The low level of ACE2 expression in the lung has been mentioned in the original text of the Introduction:

“Several studies on the tissue tropism of ACE2, the primary viral receptor, found that it was rather weakly expressed in the lung, contrary to expectations based on the typical manifestation of COVID-19 as a respiratory illness (5)...”

“These findings are further supported by our prior study characterizing ACE2 expression based on bulk and single-cell RNA-Seq datasets, which reported low levels of ACE2 expression in the lung (9).”

The relevance of ENPEP in fibroblasts of lung has been mentioned in the original text of the Discussion:

“These characteristics attributed to alveolar fibroblasts are remarkably similar to those of pericytes, defined by their localization in the basal lamina directly beneath the endothelium and a characteristic morphology consisting of cytoplasmic processes (67), suggesting that these alveolar fibroblasts might play similar roles in the lung parenchyma, guiding vessel development around the alveoli, with APA playing a regulatory role.”

We have added the following new text to the Discussion:

“Importantly, as we and others have shown previously, expression of ACE2 in the lung is very limited, with arterial vascular endothelial cells of the lung showing the greatest proportion of ACE2 expression at just 2.55% of cells (Barker and Parkkila 2020; Adams et al. 2020). Our current analysis of scRNA-Seq data from the Human Protein Atlas data likewise shows limited expression of ACE2 in the lung, where the highest distribution of expression is observed in a type 2 alveolar cell cluster, with just 3.6% of cells expressing. By comparison, in the same dataset, ENPEP is expressed in 6.5% of lung fibroblasts and 6.5% of lung smooth muscle cells. This low level of expression of ACE2 in the primary organ of pathology in COVID-19 infection points to other functional causes of disease severity to which ENPEP may be a contributor, namely the RAS/RAAS.”

Reviewer 1 Comment 4

2. Methodological Clarity:

The bioinformatic methods used in this study are comprehensive; however, the manuscript could benefit from a more detailed explanation of certain steps, especially for readers less familiar with bioinformatics. For instance, more clarity on the thresholds used for coexpression analysis and gene ontology enrichment would be helpful. It is also important to include justification for selecting specific datasets and cutoff values in the analyses.

Author Response 4

We thank the reviewer for pointing this out and we endeavor to make the approaches more comprehensible to non-experts in bioinformatics. Thresholds are often set at higher levels to narrow results to specific biological functions (e.g., cell adhesion) or set more loosely to allow determination of broader biological programs (e.g., cell cycle). Depending on the dataset, thresholds can be set based on upon over-abundance or deficit of correlating elements. However, commonly a Speaman/Pearson value of 0.7–0.8 is used to denote “strong” correlation. An often cited (PMID: 23638278; 7,700+ citations) article on the topic sets the following delineations: moderate (0.50–0.70), high (0.70–0.90), very high (0.90–1.00). 

As a result, we have added a reference to this article at the mention of the correlation thresholds used, in the Methods and Results sections, respectively:

“Genes satisfying the high correlation threshold cutoff (≥0.7) (Mukaka 2012) and a...”

“Using expression values from all tissues in the GTEx database and a high correlation cutoff of 0.7 (Mukaka 2012),...”

Reviewer 1 Comment 5

3. Single-cell RNA-Seq Analysis:

The single-cell RNA-Seq data provides valuable insights, particularly regarding ENPEP expression in specific cell types like pericytes and enterocytes. However, it is recommended that the authors further elaborate on the potential functional implications of ENPEP expression in these cells, particularly in the context of viral entry and infection.

Author Response 5

Thank you for the comment emphasizing the need for clarity on the functional implications regarding the expression pattern of ENPEP. In the Discussion we examined several possible tissue roles for ENPEP in enterocytes and pericytes, and now have added discussion in the context of viral entry and infection of enterocytes and pericytes. We also summarized some of the main ideas near the end of the discussion.

We have modified/added the following new text to the Discussion (italics indicates old text when new text is a modification of an existing sentence):

“While It appears that the expression profiles of ACE2 and ENPEP align in vasculature and particularly pericytes, we found that many of the known SARS-CoV-2 cofactors had weak correlations with ACE2 in blood vessels.“

“The absence of ENPEP epithelial cell expression outside the small intestine also limits the areas where APA could encounter an outside pathogen, making it unlikely to have a role in the initial target cells of SARS-CoV-2: small intestine enterocytes appear to be the only surface cell type exhibiting significant ENPEP expression. While SARS-CoV-2 does directly infect enterocytes (Lehmann et al. 2021), COVID-19 is nevertheless primarily a respiratory illness primarily transmitted by droplets or aerosols (Hu et al. 2021). While more study is required on the interplay of ACE2 and ENPEP in the intestine if APA is found to be a SARS-CoV-2 cofactor, the most interesting interactions between APA, ACE2, and SARS-CoV-2 are likely found in blood vessels, noting the overlapping expression profiles in pericytes (Schlingemann et al. 1996; He et al. 2020; Barker and Parkkila 2020) and the significance of the vascular system in COVID-19 (Siddiqi et al. 2020; Rysz et al. 2021).”

“In summary, both APA and ACE2 appear to be present in systemic vasculature and pericytes in particular — a cell type implicated to have a significant role in COVID-19 (He et al. 2020). If SARS-CoV-2 utilizes APA as a cofactor to infect enterocytes, this could contribute to systemic inflammation in COVID-19 and influence other pericyte-mediated functions such as capillary and blood-brain-barrier permeability (Armulik et al. 2005; Balabanov and Dore-Duffy 1998). ENPEP and ACE2 both have important roles in the RAAS, with RAAS dysregulation suggested as an important part of COVID-19 pathophysiology (Rysz et al. 2021).”

“If SARS-CoV-2 interacts with APA it could help explain the vascular effects of COVID-19 since pericytes are a systemic reservoir of cells in which APA and ACE2 are more strongly correlated than many of the currently known cofactors. This pericyte-targeted infection could promote the systemic vascular inflammation, capillary leakage, and blood-brain-barrier disruption discussed above.”

Reviewer 1 Comment 6

Additionally, the authors should discuss the limitations of using scRNA-Seq data, such as potential dropout effects or the challenge of distinguishing between similar cell types.

Author Response 6

We appreciate the comment by the reviewer as this had indeed not been addressed adequately. We have added a study limitations and future directions section at the end of the Discussion (see Author Response 7).

Reviewer 1 Comment 7

4. Experimental Validation:

While the manuscript provides compelling bioinformatic evidence for ENPEP as a potential cofactor, it would greatly benefit from experimental validation, as suggested by the authors. The authors should discuss the feasibility of experiments, such as knockdown or overexpression studies, to validate ENPEP's role in SARS-CoV-2 infection. If experimental validation is beyond the scope of this study, the authors should clearly indicate this as a limitation and suggest future directions for experimental follow-up.

Author Response 7

This is a very useful comment for additional considerations, as in Comment 6, we have now included together with the newly-added study limitations additional discussion on future directions.

“The study performs several analyses based on RNA-Seq data, both bulk and single-cell. Actual levels of protein and RNA expression are often loosely correlated and thus high or low values of gene expression may not fully reflect the actual biological state of the tissues analyzed. Additionally, scRNA-Seq has inherent limitations, including low sequencing depth and dropout events. Classification of cell types by gene signatures, especially for highly specialized cells or specific states, is an evolving field which leaves room for interpretation and future improvement. As this study provides an initial bioinformatics characterization, further experimental studies are merited which directly test the role of ENPEP as a potential co-factor in SARS-CoV-2 infection, and the roles of ENPEP and ACE2 in RAS/RAAS in COVID-19 pathophysiology. For the next investigatory steps, we envision co-immunoprecipitation investigation of ENPEP with the full SARS-CoV-2 proteome and RNA-Seq of ACE2-expressing airway epithelial cell lines in ENPEP knockdown, knockout, and overexpression.”

Reviewer 1 Comment 8

5. Figure Presentation:

The figures, particularly those presenting tissue-specific expression and immunohistochemistry images, are informative. However, some figures could be improved for clarity. For example, Figure 1 presents tissue-specific ENPEP expression, but it would be helpful to include additional labeling or annotations to make it more accessible to readers.

Author Response 8

Thank you for the suggestion of making Figure 1 more accessible to readers.

We have made the following modifications to the Figure 1 image:

Adjusted x-axis text for readability (size and rotation).

Removed Ensembl identifiers.

Updated y-axis title to better reflect content of plot.

Added grid lines for better interpretability.

We have made the following modifications to the Figure 1 caption (italics indicates old text when new text is a modification of an existing sentence):

“Figure 1. Tissue specific expression of ACE2 and ENPEP in 55 human tissues. Bulk RNA-Seq expression data from the GTEx dataset (v8)(45) was retrieved as TPM values and boxplot figures were generated using the Seaborn (49) and Matplotlib (49) Python libraries. Each boxplot displays the distribution of expression values for the gene-tissue intersection: the box represents the interquartile range (IQR), the lower boundary marks the 1st quartile (Q1), the upper boundary marks the 3rd quartile (Q3), and the horizontal line within the box indicates the median. The whiskers extend to the maximum and minimum values within 1.5 times the IQR”

We have made the following modification/addition to the Figure 8 caption (italics indicates old text when new text is a modification of an existing sentence):

“Each boxplot displays the distribution of expression values for the gene-tissue intersection: the box represents the interquartile range (IQR), the lower boundary marks the 1st quartile (Q1), the upper boundary marks the 3rd quartile (Q3), and the horizontal line within the box indicates the median. The whiskers extend to the maximum and minimum values within 1.5 times the IQR.”

We have made the following modification/addition to the Figure 9 caption (italics indicates old text when new text is a modification of an existing sentence):

“Each boxplot displays the distribution of expression values for the gene-tissue intersection: the box represents the interquartile range (IQR), the lower boundary marks the 1st quartile (Q1), the upper boundary marks the 3rd quartile (Q3), and the horizontal line within the box indicates the median. The whiskers extend to the maximum and minimum values within 1.5 times the IQR.”

Reviewer 1 Comment 9

The immunohistochemistry images should also be clearly labeled with magnification and staining details for better interpretation.

Author Response 9

Than

---

## [Editor Report · Decision Letter 1]

22 Nov 2024

Bioinformatic characterization of ENPEP, the gene encoding a potential cofactor for SARS-CoV-2 infection

PONE-D-24-28420R1

Dear Dr. Barker,

We’re pleased to inform you that your manuscript has been judged scientifically suitable for publication and will be formally accepted for publication once it meets all outstanding technical requirements.

Kind regards,

Cheorl-Ho Kim, Ph.D.

Academic Editor

PLOS ONE

Additional Editor Comments (optional):

Dear Dr Barker,

Thank you for your submission of your study and revision to the PLOS ONE.

I have checked your revision and would appreciate you for your appropriate responses to our external reviewers.

I don't forward your revision to our original reviewers as I have find your revision is relevant.

Now I am very pleased to inform that your revision is acceptable for publication in Plos One.

Thank you

Sincerely

Cheorl-Ho Kim PhD Professor

Editor
---

## [Editor Report · Acceptance letter]

30 Nov 2024

PONE-D-24-28420R1 

PLOS ONE

Dear Dr. Barker, 

I'm pleased to inform you that your manuscript has been deemed suitable for publication in PLOS ONE. Congratulations! Your manuscript is now being handed over to our production team.

Kind regards, 

on behalf of

Professor Cheorl-Ho Kim 

Academic Editor

PLOS ONE